# Pathologist-like explainable AI for interpretable Gleason grading in prostate cancer

The aggressiveness of prostate cancer is primarily assessed from histopathological data using the Gleason scoring system. Conventional artificial intelligence (AI) approaches can predict Gleason scores, but often lack explainability, which may limit clinical acceptance. Here, we present an alternative, inherently explainable AI that circumvents the need for post-hoc explainability methods. The model was trained on 1,015 tissue microarray core images, annotated with detailed pattern descriptions by 54 international pathologists following standardized guidelines. It uses pathologist-defined terminology and was trained using soft labels to capture data uncertainty. This approach enables robust Gleason pattern segmentation despite high interobserver variability. The model achieved comparable or superior performance to direct Gleason pattern segmentation (Dice score: $0.713_{\pm 0.003}$ vs. $0.691_{\pm 0.010}$) while providing interpretable outputs. We release this dataset to encourage further research on segmentation in medical tasks with high subjectivity and to deepen insights into pathologists' reasoning.

Prostate cancer is a major health issue, affecting approximately 5 million men globally, with around 1.5 million new cases reported in 2020[1]. The Gleason grading system, developed by Donald Gleason in 1974[2] and most recently discussed and updated by the International Society of Urology Pathology (ISUP) and by the Genitourinary Pathology Society (GUPS) in 2019[3,4], remains the primary method for assessing tumor aggressiveness and prognosis in patients with prostate cancer[5].

For Gleason scoring in the context of primary diagnosis, pathologists assess histological architectural features such as gland shape and size based on tumor biopsy samples and assign Gleason patterns ranging from 1 (resembling gland-like structures) to 5 (resembling least gland-like structures). Gleason patterns 1 and 2 were merged with pattern 3 in later modifications of the system[6], therefore, the Gleason score, which in tissue microarrays (TMAs) is quantified as a sum of the most predominant and the highest Gleason pattern, ranges from 6 (3 + 3) to 10 (5 + 5). Higher scores indicate more aggressive tumors[7]. Despite its widespread use, however, the Gleason system has limitations, including sampling bias and subjective assessment of tumor architecture, resulting in significant interobserver variability[8].

Multiple studies have shown that artificial intelligence (AI)-based image analysis has the potential to assist pathologists in Gleason grading, potentially matching or exceeding human accuracy[9–12]. Developing robust AI models for this task requires large datasets with expert annotations. For Gleason grading, datasets such as the Gleason19 Challenge[13,14] and the PANDA Challenge[11,12] are openly available; however, in most cases the available annotations indicate the area of patterns relevant to the final scoring or merely the Gleason score, without providing an explanation of the specific histological criteria behind the decisions. Consequently, the typical approach to Gleason grading with AI involves end-to-end models that predict Gleason patterns or the score directly from the images. Although these models can achieve high accuracy, their decision-making process lacks transparency, which may present a barrier to clinical adoption[15,16], particularly in light of patients' right to explanation[17]. Especially in fields such as Gleason grading, where there is a significant subjectivity in the assessment[8], the demand for clear and explainable explanations for AI-assisted diagnostic systems is high.

In order to overcome interpretability issues for neural networks, post-hoc explainability techniques such as CAM or Grad-CAM[18], LRP[19],

e-mail: titus.brinker@dkfz.de

and LIME[20] have been developed that highlight regions of interest relevant to the decision made. These heatmaps aim to provide visual explanations for AI decisions, for example by indicating pixels that have a high influence on the predicted outcome[21,22]. However, these methods often provide pseudo-explainability with vague morphological correlates. Interpreting the results requires specialized expertise[23], and it has been shown that using these approaches carries a high risk of confirmation bias[24,25], a cognitive tendency whereby individuals favor evidence that confirms their pre-existing hypotheses and beliefs. Additionally, such indicated regions may not always correspond to the actual causative regions of the cancer patterns[26,27], but might instead show unwanted statistical correlations learned by the neural network - a crucial factor that is rarely addressed. As pathologists prefer simple, visual explanations that are grounded in morphology and reflect their way of thinking, an inherently explainable approach to AI with clear and intuitive explanations is needed[28].

To address these significant limitations of traditional algorithm development, we propose the use of a concept-bottleneck-like[29] U-Net[30] architecture to develop a pathologist-like, inherently explainable AI system (GleasonXAI), as presented in Fig. 1. For the development, we compile and open-source one of the largest datasets of annotations localizing explanations for Gleason patterns in TMA core images. The resulting GleasonXAI offers interpretability for AI-assisted segmentation of Gleason pattern by directly recognizing and delineating pre-defined histological features, which are associated with a textual explanation using terminology common to pathologists and rooted in GUPS and ISUP recommendations.

Utilizing the approach, we developed to train and evaluate models with soft labels, we capture the intrinsic uncertainty in the training data, thereby providing promising Gleason pattern segmentation in spite of high interobserver variability, and exceeding the performance of traditional approaches directly trained to predict Gleason patterns.

## Results

### Pathologist characteristics
Between March 2023 and October 2023, an international team of 54 pathologists from ten countries participated in the study, with the majority of participants from Germany (22) and the USA (18). Among the participants, 47 were responsible for explanatory annotations, six for Gleason grade annotations, and one for creating the initial terminology, which was later reviewed and adapted in a panel of nine of the participants (see Methods section). The pathologists had a median of 15 years of clinical experience in pathology, with individual experience ranging from one to 35 years. Notably, 28 of these 54 annotators had extensive experience, defined as 15 years or more. In their clinical practice, the participating pathologists signed out a median of 15 prostate cancer cases per week, with the average individual prostate cancer caseloads ranging from fewer than ten to 75 patient cases weekly.

### Dataset characteristics
The annotated dataset generated in this study comprised 1015 TMA core images. The retrospective collected images were sourced from three distinct datasets, each created by a different institution. The annotations consisted of areas to which explanations describing histological patterns were assigned. Explanations could be mapped to one of the three Gleason patterns or further divided into more detailed sub-explanations, which describe more specific subgroups of the histological features defined by their parent explanation. For additional information, please refer to the Methods section.

For each of the Gleason patterns, there were a considerable number of images containing associated annotations. Specifically, 55.76% of the images contained annotations for Gleason pattern 3 (566/1015), 74.48% for Gleason pattern 4 (756/1015), and 32.32% for Gleason pattern 5 (328/1015) (see Fig. 2a). When analyzing the segmentation masks on both the explanation and sub-explanation level,

the number of images containing the classes at least once exhibited a higher variation, with values ranging from 57 to 729 for explanations and zero to 526 for sub-explanations (see Fig. 2b, c). However, the broader, classical explanations yielded a more balanced distribution, with fewer small classes.

### Agreement between pathologists varies depending on histopathologic pattern
Understanding the interobserver variability in the ground truth for Gleason grading was crucial for improving the reliability of our classifier, as inconsistent annotations can significantly impact model performance.

In our dataset, the images were accompanied by Gleason score information (see Methods section), generated for each core by a consensus between one to six pathologists, to provide guidance to the annotators. Pathologists were, however, encouraged to use explanations of different Gleason patterns, if they disagreed. Comparing the accompanying grade information with the annotated Gleason patterns (see Fig. 3a), the grades largely aligned. The majority of discrepancies occurred at the boundaries between Gleason patterns 3 and 4, and Gleason patterns 4 and 5 - an expected observation, as borderline cases are a known source of interobserver variability. The high agreement is also reflected in the Fleiss' kappa[31] values, which ranged from 0.23 to 1.00 within the annotator groups when identifying Gleason patterns (see Fig. 3b, top), indicating *fair* to *perfect agreement* according to Landis and Koch[32].

Consensus on the specific histological patterns, however, and consequently the appropriate sub-explanations and explanations, was less frequent. The extent of this variability differs depending on the specific explanation under consideration, with Fleiss' Kappa values of the groups ranging from −0.05 to 0.86 for the explanations.

An analysis of the distribution of the images across the number of annotators that agreed on the presence of each explanation (see Fig. 3b, left) revealed high levels of agreement for certain histological features, such as *poorly formed glands* and *individual glands*. Specifically, 76.13% (555/729) of the images annotated with poorly formed glands and 80.81% (445/563) with individual glands by at least one annotator reached at least two-rater agreement. This is further supported by their respective mean Fleiss' kappa values of $0.50_{\pm 0.23}$ and $0.61_{\pm 0.20}$ (see Fig. 3b, right), indicating *moderate* to *substantial* agreement.

Conversely, there are also explanations, such as *glomeruloid glands* and *single cells*, where it was rare for a second or third annotator to agree (see Fig. 3b, left). Subgroup analyses identified particularly pronounced interobserver variability for the explanations of *single cells* and *compressed glands*, with Fleiss' kappa values of 0.145 and 0.180, respectively (see Supplementary Table 4). The explanations of *glomeruloid glands* and *comedonecrosis* on the other hand exhibited large variance in Fleiss' kappa values across the annotator groups denoted by values ranging from −0.031 to 0.796 and −0.052 to 0.852, respectively. Notably, with the exception of *compressed glands*, these explanations were the rarest annotated classes (see Fig. 2).

The agreement on the sub-explanation was notably lower, with Fleiss' kappa values ranging from −0.22 to 0.85 in the labels between the groups, as illustrated by the predominantly *slight agreement* shown in Supplementary Fig. 1 for each label. As interobserver agreements for most histologic patterns, which are equivalent to our explanations, are reported to be *fair* or *moderate*[33–35], it is to be expected that agreement on even finer details of the patterns will be lower. These results indicated considerable noise in the identification of sub-explanations, confirming the necessity of using the explanations that consolidate the detailed sub-explanations into broader, medically coherent categories. This step was crucial to reduce variability and to ensure more reliable training of our classifier.

## a) Gleason XAI

## b) Annotation Process

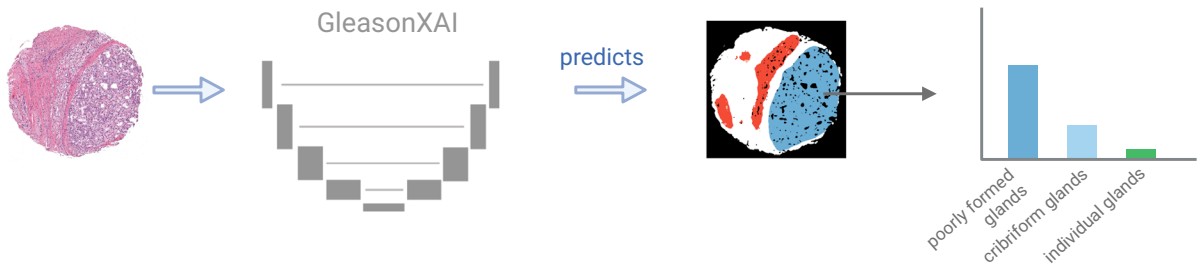

**Fig. 1 | Overview. a** We developed GleasonXAI, a U-Net model that predicts the presence of histological features closely aligned with the pathologists' consensus. Due to training with soft labels, the predicted distribution often reflects the agreement of the annotators. **b** In the annotation process, up to six pathologists evaluated the TMA core images, identifying areas for each Gleason pattern, which were then merged using the simultaneous truth and performance level estimation (STAPLE) algorithm. Subsequently, three pathologists independently annotated histologic patterns based on a predefined ontology. We compared training on two labeling approaches: soft and hard labels. In the soft label approach, each pixel is represented as a distribution across the annotated classes, while the hard label method assigns a class to each pixel through majority voting. Further details on post- and pre-processing, such as the masking of background pixels, can be found in the Methods section. Created in bioRender.com[64].

Further details on the Fleiss' kappa values and their bootstrapped confidence intervals can be found in Supplementary Tables 4 to 9.

**Pixelwise agreement between raters is lower in minority classes**
Since the AI was tasked with learning the localization of the explanations, the annotators' agreement at the pixel level was crucial. As their annotations served as the predictive targets for the AI, the level of interobserver agreement induced an upper limit on the performance the AI could reach.

A similar pattern of decreasing annotator agreement with increasing explanation detail was observed when analyzing the number of pixels with a unique majority vote. Of the 58.12% of pixels constituting

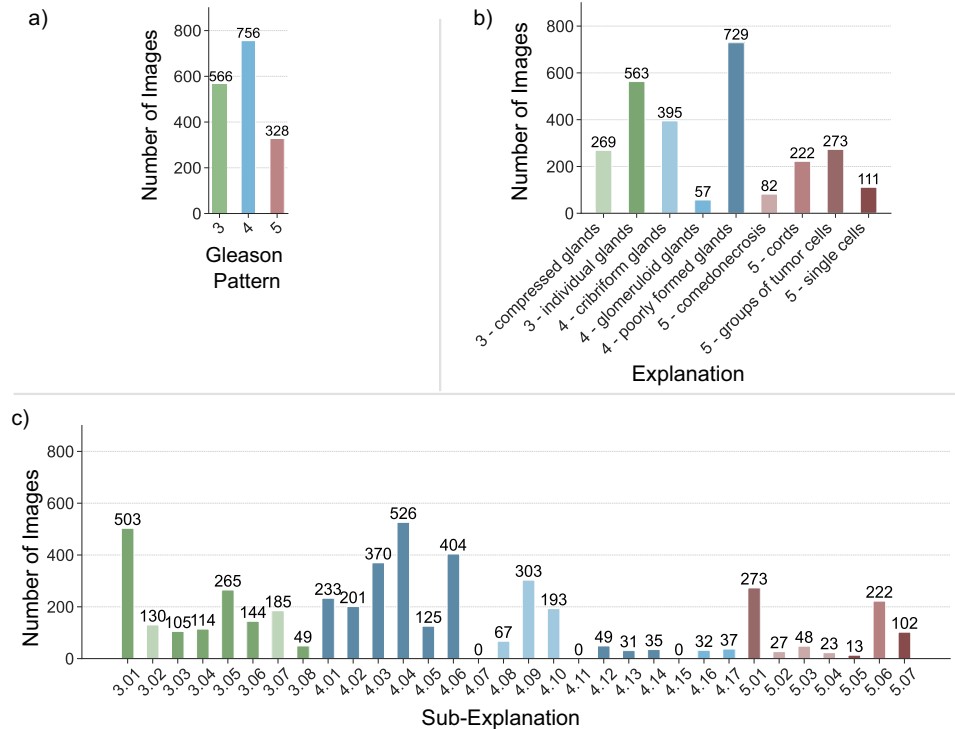

**Fig. 2 | Class distribution.** Number of TMA core images with at least one occurrence of **a** the specified Gleason pattern, **b** the specified explanation, and **c** the specified sub-explanation. *Benign tissue* is not included, as it is present in all images. A mapping of sub-explanation numbers to text and is available in Supplementary Tables 1 to 3.

The mapping of the explanations to their long version is available in the Methods. Colors of sub-explanations in (**c**) map to the colors of their parent explanation in (**b**). All green colors map to Gleason pattern 3, blue colors to Gleason pattern 4, and red colors to Gleason pattern 5. Source data are provided as a Source Data file.

the foreground of the dataset, 97.54% could be assigned a unique majority class when evaluating the Gleason patterns (36.23% with two raters and 61.30% with three raters agreement). However, at the explanation level, this dropped to 86.41% (41.07% with two raters and 45.35% with three raters agreement, respectively), and further to 67.76% at the sub-explanations level (37.80% and 29.96%, respectively), indicating a considerable proportion of pixels with a high annotation uncertainty.

It is worth noting that the classes with the lowest number of annotated pixels (see Fig. 4b) were also precisely the classes where annotators demonstrated the least consensus. As illustrated in Fig. 4a), this was particularly evident for the explanations of *comedonecrosis*, *single cells*, *glomeruloid glands*, and *compressed glands*, where 88.46% to 94.96% of all annotated pixels were annotated by a single rater.

The lower number of pixels with a unique majority vote indicated that the agreement at the pixel level was weaker compared to the image level. Consequently, it could be inferred that Fleiss' kappa at image-level represented an upper bound on the overall agreement. As demonstrated in Fig. 3c), the agreement on the existence of an explanation does not necessarily concur with the agreement on the location, as the outline and granularity of the annotation differs between the pathologists.

### Model development and evaluation
**Soft labels improve model performance by considering annotator uncertainty.** To develop a pathologist-like, inherently explainable AI system for Gleason pattern segmentation (Gleason XAI), we selected a soft label approach by treating the different annotations from different annotators over the pixels as probability distributions. This approach accounted for the high interobserver disagreement stemming from the reviewers' annotation uncertainty. Preserving all annotations instead of merging them with a traditionally used majority vote ensured that the AI system could also reflect the nuances of expert judgment. For comparison, we also included classical hard label approaches, using the majority votes of our international expert team of pathologists.

For both approaches, we compared our models trained on the training data with different loss functions (see Methods section). Specifically, for the soft label approach, we used the cross-entropy loss and our custom SoftDiceLoss, while for the hard label approach, we employed the original Dice loss and cross-entropy loss (see Fig. 5). All approaches were compared using the Macro SoftDice based on the loss function of Wang et al.[36], $L_1$-norm, Dice and Macro Dice metrics on a holdout test set. Due to the large interobserver disagreement and class imbalance present in the sub-explanations, we trained our models on the broader, medically coherent explanation level of our ontology (see Methods section, Fig. 8). A comparison of all methods, when trained on the sub-explanations, as well as the full numerical results can be found in Supplementary Fig. 2 and Supplementary Tables 11 and 12.

For most models, the metrics showed a decline from the evaluation on Gleason patterns to the explanations. This deterioration could be attributed to the increased number of classes, greater class imbalance in the segmentation task, and higher inter-rater variability, particularly affecting minority classes. Due to the high class imbalance in the explanations, this trend was especially noticeable in the class-balanced metrics (i.e, Macro Dice and Macro SoftDice).

Training on majority-voting-based explanation labels, as opposed to soft labels, resulted in a slight decrease in segmentation quality when evaluated on the explanations, especially when comparing the soft-label approaches with the cross-entropy loss. The difference became much more pronounced when the predictions on the explanations were mapped to the higher-level Gleason patterns, where models trained on majority-voted labels performed worse than those trained using soft labels. Trained on the Gleason patterns, the hard and soft label approaches performed on par, with the cross-entropy loss on the soft labels achieving the highest Dice and Macro Dice. Overall, the soft label-based approaches consistently demonstrated superior performance in terms of the segmentation metrics on the test data compared to hard label approaches, when trained on the explanations.

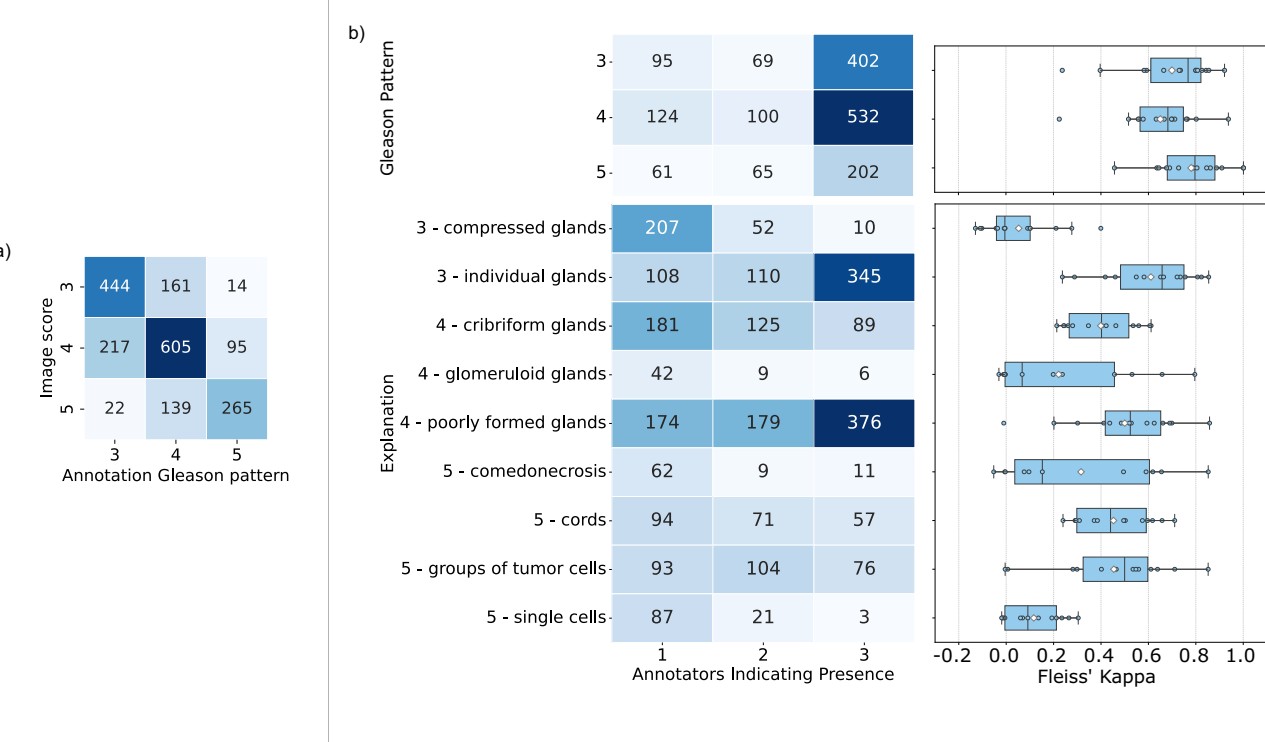

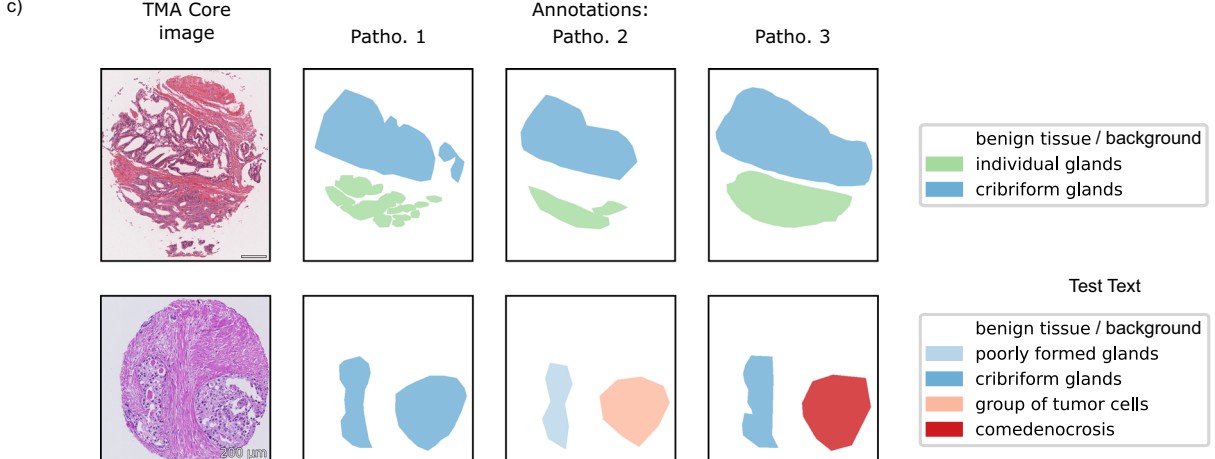

**Fig. 3 | Agreement of annotators for explanations on the image-level.**
**a** Confusion matrix between the Gleason score presented to the annotators and Gleason pattern of the applied explanations in the images (Each Gleason pattern was counted only once per image, regardless of number of agreeing pathologists), and **b** heatmap containing the number of TMA core images in which n out of the three annotators indicated the presence of the Gleason pattern (left, top) and explanations (left, bottom), and the resulting Fleiss' kappa within groups of three raters (on the right) as boxplot. In the boxplot, dots represent the Fleiss' kappa value of a group of $n = 3$ annotators. Boxes represent the inter-quartile range, with the centre line marking the median, the white diamonds mark the mean value, and whiskers extend to the minimum and maximum within 1.5 of the inter-quartile range. For most Gleason patterns and explanations, the Fleiss' kappa values of 14 groups of annotators are included. As not every group used all categories, the number of groups taken into account is reduced for *glomeruloid glands* (13 groups), *single cells* (13 groups), and *comedonecrosis* (11 groups). Precise numerical values can be found in Supplementary Tables 8 to 10, and the figure for sub-explanations in Supplementary Fig. 1. The mapping of the explanations to their long version is available in the Methods. Exemplary differences in the consensus of the three annotators are shown in (**c**) with an example for high agreement in the first row and an example for labels with low class agreement in the bottom row. The scale bar corresponds to 200 μm. Source data are provided as a Source Data file.

Importantly, we were able to preserve the segmentation quality of the Gleason patterns even when training on the explanations. Models using the SoftDiceLoss, trained on the explanations but evaluated on the Gleason patterns, performed just as well as those trained directly on the Gleason patterns. This was reassuring, as it demonstrated that the inherent interpretability of our method did not come at the cost of reduced segmentation performance for the clinical task.

Similarly to the segmentation performance, models trained on explanations with our custom SoftDiceLoss exhibited better calibration to the pathologists' annotation distribution compared to the cross-entropy models trained with soft labels, as evidenced by a lower $L_1$-norm (see Fig. 5).

Since the models trained with SoftDiceLoss on the explanations performed best across most metrics – except for the $L_1$-norm on the explanations – on both the explanations and Gleason patterns, we will

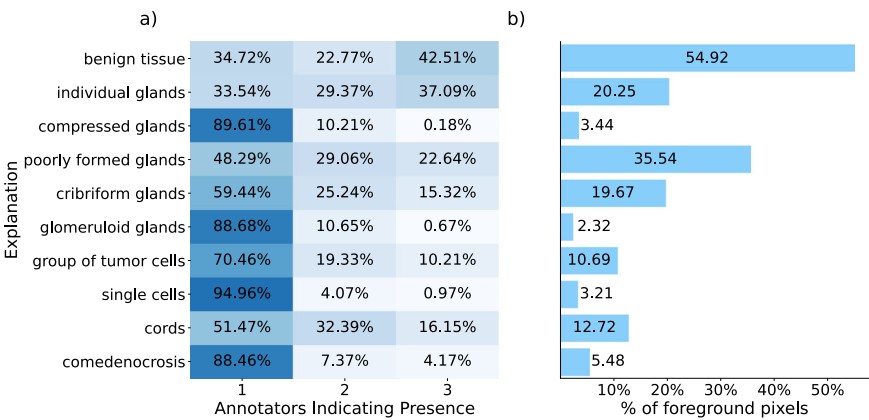

**Fig. 4 | Agreement of annotators for explanations on pixel-level.** Agreement demonstrated by **a** the proportion of pixels annotated for a given class by at least one annotator, stratified by the number of annotators indicating the presence of the explanation and **b** the percentage of foreground pixels annotated with an explanation by at least one annotator. The mapping of the explanations to their long version is available in the Methods.

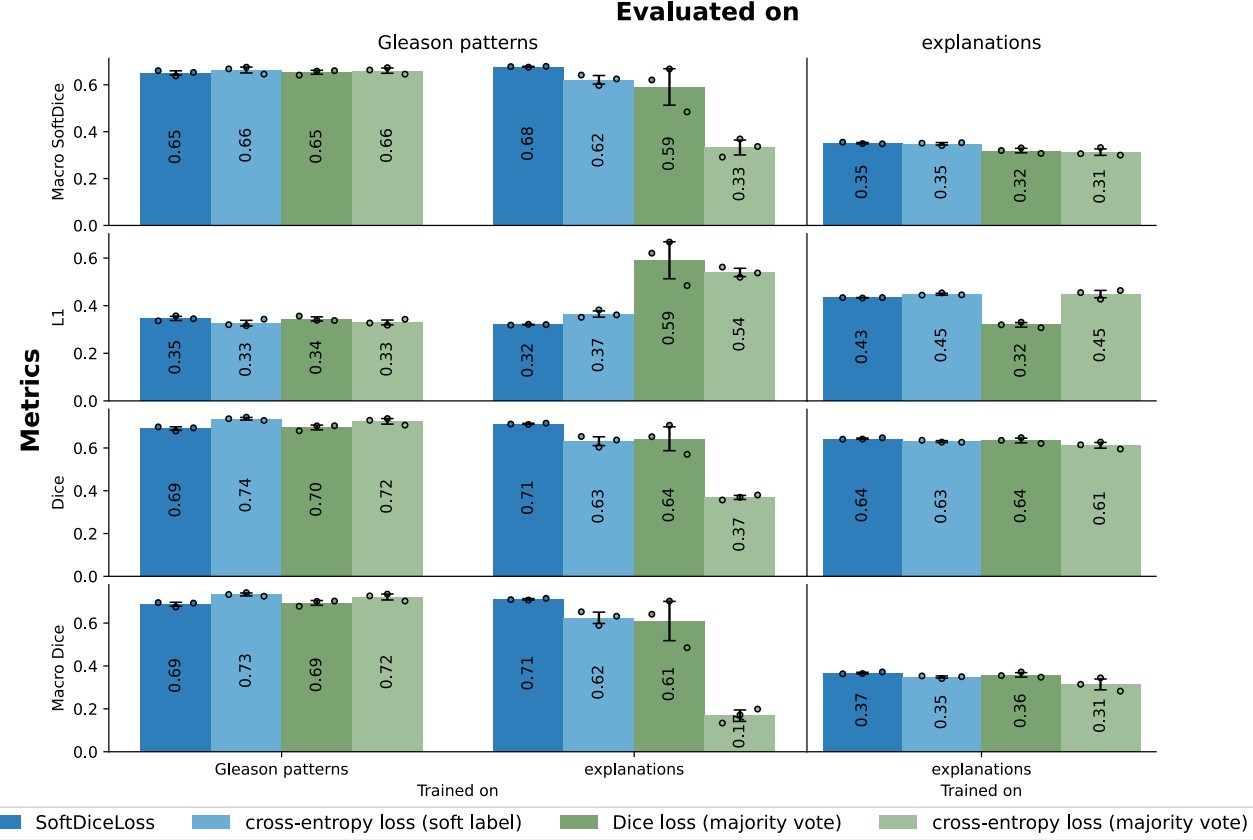

**Fig. 5 | Results.** Results for our models trained with different loss functions, evaluated on the Gleason patterns and the corresponding explanations. Using our ontology, we mapped the labels upwards, allowing a comparison between the models trained on the explanations with those directly trained on the Gleason patterns. The bar plots display both the mean and the standard deviation of three models trained with different seeds but the same hyperparameters, with the mean values additionally indicated within the bars. The result of the $n=3$ technical replicates are indicated as dots. The green bar charts represent metrics for the hard label approaches, while the blue bars correspond to the soft label approaches. For the Dice metrics, higher values indicate better performance, while for the L1-norm, lower values are preferable. Source data are provided as a Source Data file.

use these models for the remainder of the analysis and define them as our GleasonXAI models. Further discussion on the calibration can be found in the Supplementary Discussion.

**GleasonXAI strongly aligns with pathologists.** To further analyze the performance capabilities of our GleasonXAI models, we examined the distributions of predictions and the confusion matrices of the models (see Fig. 6) on the test set. We combined the results of the three models by averaging the predictive distribution and confusion matrices.

Our GleasonXAI models predicted the majority of classes in the dataset with high reliability, often closely matching the prediction frequency of the annotations. Classes that were rarely annotated were predicted more frequently than they were annotated, and were also assigned a greater probability mass (see Fig. 6a). However, for the

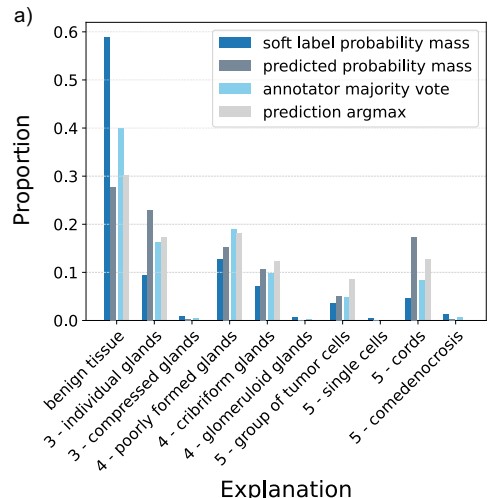

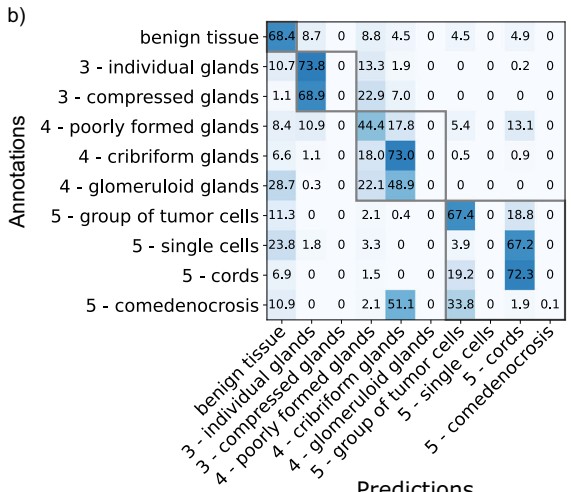

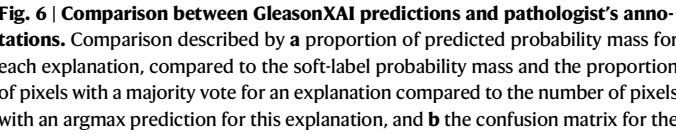

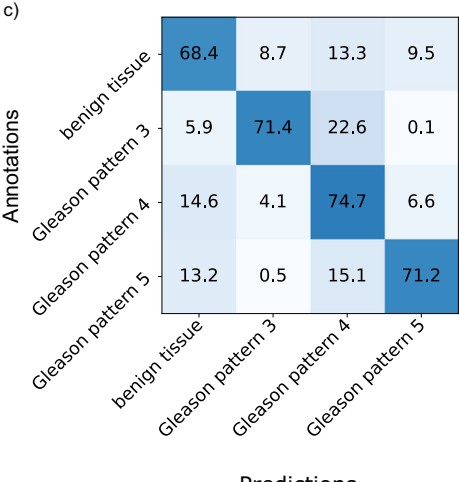

**Fig. 6 | Comparison between GleasonXAI predictions and pathologist's annotations.** Comparison described by **a** proportion of predicted probability mass for each explanation, compared to the soft-label probability mass and the proportion of pixels with a majority vote for an explanation compared to the number of pixels with an argmax prediction for this explanation, and **b** the confusion matrix for the argmax prediction and the majority label, presented in percentages of pixels. The gray boxes highlight the explanations corresponding to a common Gleason pattern. For a more comprehensible representation, **c** illustrates the confusion matrix in percentage when the explanations were mapped to Gleason patterns. The mapping of the explanations to their long version is available in the Methods.

rarest explanations − such as *glomeruloid glands*, *single cells*, and *presence of comedonecrosis* − the methods were not able to produce predictions with high certainty, likely due to their rarity in the training data and due to these classes rarely being the majority vote (see Fig. 4). This hypothesis is further supported by the observation that the non-predicted explanations are often most frequently confused with classes that frequently overlap within the pathologists' annotations, i.e. 55.23% of the pixels annotated with *comedonecrosis* were also labeled with *solid groups of tumor cells* by at least one annotator. However, probability mass was assigned to these rare classes, which indicates that they are not fully unrecognized, but rather their predictions are aligned with the pathologist annotations.

The confusion matrix in Fig. 6b illustrates the annotations versus the predictions, revealing minimal confusion between explanations of different Gleason patterns, as indicated by the gray boxes, which underscored the strong performance of our models trained for Gleason pattern segmentation. Notably, explanations of Gleason patterns 3 and 5 were rarely misclassified as one another. As demonstrated in Fig. 6c, misclassifications predominantly occurred between adjacent Gleason patterns. This outcome was expected, as many cases could be

medically categorized as falling between the Gleason pattern stages, and aligned with the pathologists' grading discrepancies shown in Fig. 3a, where they also mostly differed between adjacent classes. Overall, all Gleason patterns were detected with comparable reliability, regardless of the number of belonging explanations, share of the dataset or the segmentation quality of the underlying explanations.

The class that was most frequently falsely classified was *benign tissue*, which we hypothesize is likely due to label uncertainty in the border regions of the annotations. This uncertainty might have arisen from the inherent difficulty in precisely determining annotation boundaries, as demonstrated in Fig. 3c. Despite this, most of the predicted classes were predicted with high accuracy, with the lowest accuracy observed for the Gleason pattern 4 explanation of *poorly formed glands* at 44.4 % ( ± 1.18% SD) and the highest for the explanation of *individual glands* at 73.8 % (±2.66% SD).

**GleasonXAI generates detailed segmentation maps.** As segmentation maps that achieve high Dice scores can still exhibit unwanted properties like visual artifact or clutter[37], we qualitatively verified the correctness of our segmentation maps by visualizing them (see Fig. 7).

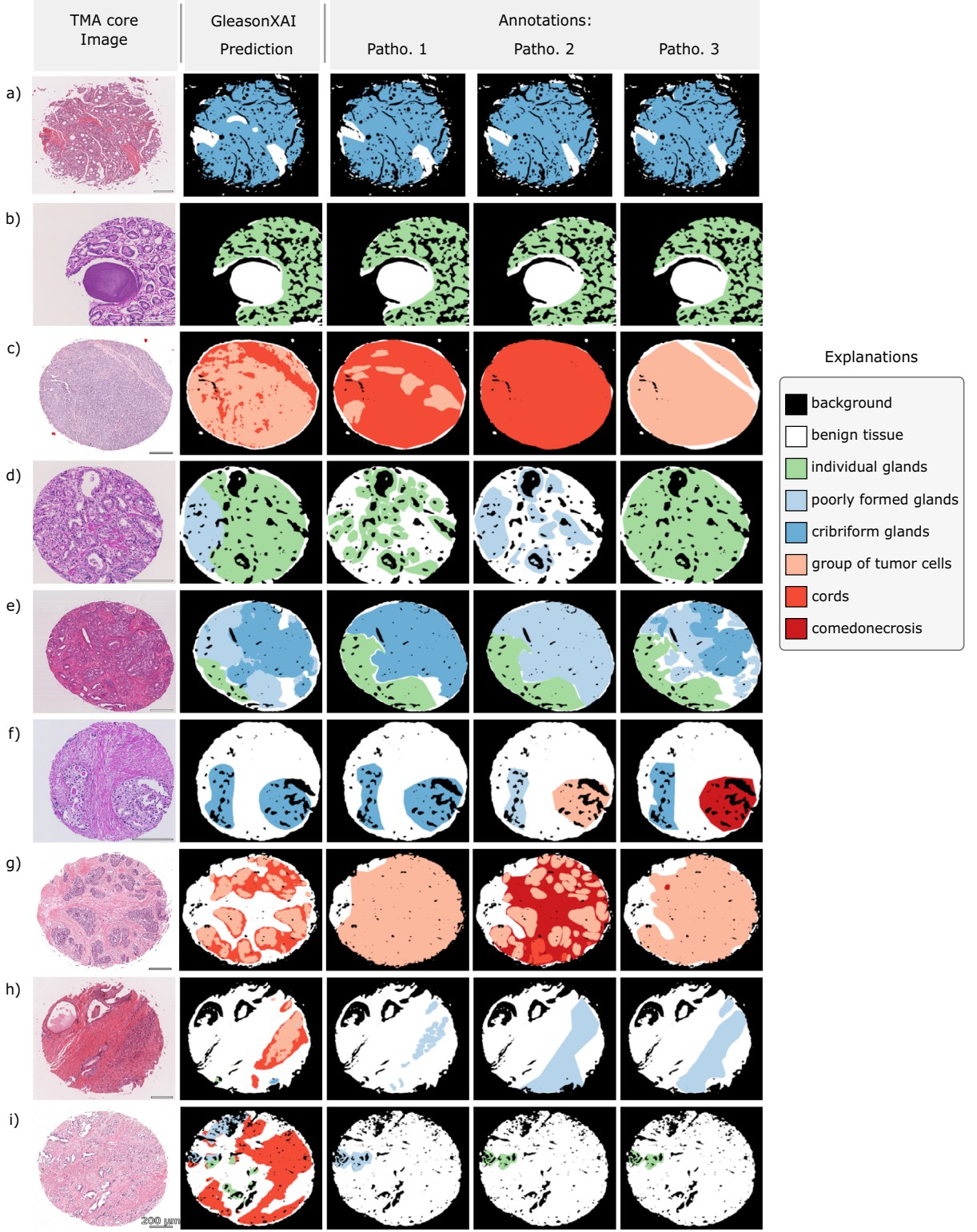

**Fig. 7 | Result Visualization.** Visualization of examples of segmentation results for the GleasonXAI model compared to the three pathologists' annotations. The segmentation images depict the argmax of the per-pixel distribution for the predictions of the model. **a**, **b** showcase examples of high agreement between the three annotators and the model. **c**–**g** highlight cases with greater disagreement among the three annotators, where the segmentation maps of the model often fell between the annotators' interpretations, reflecting the training objective of our soft-label approach. **h**, **i** illustrate instances of strong disagreement between the model and the annotators. Green labels belong to Gleason pattern 3, blue to Gleason pattern 4 and red to Gleason pattern 5. The mapping of the explanations to their long version is available in the Methods. The scale bar corresponds to 200 μm.

For these visualizations of the test data, we averaged the predictions of the three GleasonXAI models. A further analysis of the structure of the predicted features is available in the Supplementary Fig. 5.

The models generally aligned well with the provided annotations, often producing segmentation maps that integrate elements from each individual annotation. This alignment was not restricted to the majority classes; the model also effectively captured fine details of less prevalent classes, even when there was no consensus among annotators on the presence of certain patterns. Notably, our model frequently generated more detailed segmentation masks than those in the reference testset (see Fig. 7g), accurately predicting smaller details that were not annotated by multiple pathologists, confirming the high quality of our segmentations. These findings were encouraging as the model was able to also capture fine structural details and was not distracted by more coarse annotations.

## Discussion

The development of trustworthy and explainable machine learning models is crucial for their adoption in clinical practice[38]. However, previous studies have focused on detecting Gleason patterns using an end-to-end AI approach. This provides limited explainability, if any, through the use of post-hoc methods, which leaves room for potential interpretation and confirmation biases[27]. This study introduces a pathologist-like and inherently explainable model for the segmentation of Gleason patterns, trained directly on a uniquely curated dataset of expert-annotated morphological concepts. In this approach, concept-level annotations from a large, international panel of 54 pathologists that captures the full spectrum of ISUP-conformant Gleason patterns (3 + 3 to 5 + 5) are applied. In contrast to prior work, which typically addresses a limited subset of patterns or relies on labels without explicit interpretability[39], our model is grounded in clinically meaningful morphological categories that align with diagnostic practice. In addition, we publicly release a large annotated dataset of 1015 prostate cancer TMA core images along with localized, concept-based explanations.

By utilizing soft-label loss functions, such as our SoftDiceLoss, along with tailored assessment metrics, we were able to successfully train an AI system on a segmentation task characterized by substantial interobserver variability. The resulting AI is inherently explainable, directly providing explanations for tissue characteristics in accordance with the WHO/ISUP guidelines[6]. It demonstrates reliable performance across all but the rarest classes, which simultaneously exhibited the highest discordance (e.g., *comedonecrosis, glomeruloid glands* and *single cells*), often annotated only by a minority of the raters.

Remarkably, these results were achieved without any loss in the segmentation quality for Gleason patterns compared to conventional methods trained directly on them (see Fig. 5, *mean Dice*$_{\pm SD}$: $0.713_{\pm 0.003}$ vs. $0.691_{\pm 0.010}$). This is particularly encouraging, as inherently explainable XAI methods often suffer from an unfavorable performance-interpretability trade-off[40]. By treating the per-pixel annotations of multiple pathologists as soft-labels, rather than relying on high-variance, majority-voted labels, we were able to improve segmentation performance compared to hard label-based approaches, while also preserving and respecting the inherent uncertainty and ambiguity in Gleason pattern explanations.

We hypothesize that the benefits of using soft labels become more pronounced in scenarios with greater class imbalances and increased label uncertainty. This is especially visible when directly comparing the Macro Dice scores of cross-entropy models trained on soft labels (*mean*$_{\pm SD}$: $0.625_{\pm 0.032}$) against those trained on majority-voted explanations labels (*mean*$_{\pm SD}$: $0.168_{\pm 0.033}$), when evaluated on the Gleason patterns. The considerable interobserver disagreement resulted in substantial sample variance for the majority-voted labels, contributing to label-noise, while additionally requiring the exclusion of 13.59% of all foreground pixels for the explanations (see Fig. 4). By utilizing soft labels, we are able to incorporate all annotations – including minority opinions or classes that would otherwise be discarded in majority voting, or cases with multiple medically plausible annotations. This approach allows us to retain every pixel and provides an estimated annotation confidence, resulting in more conservative and distributed predictions and better predictions for minority classes.

Consequently, when training on Gleason patterns, the soft labels and the SoftDiceLoss did not provide a great advantage over the hard label-based approaches. Due to the relatively high level of agreement among pathologists on Gleason pattern level (see Fig. 3a), many soft labels closely match with the majority-voted labels, and the class distribution is more balanced. This observation aligns with recent literature[41], which suggests that label smoothing – a label augmentation technique that produces distributional labels – is particularly beneficial in settings characterized by high label noise and class imbalance.

GleasonXAI was not able to predict the rarest of classes in a majority vote evaluation (see Fig. 6), among them the morphological finding of *comedonecrosis*, which is an important histologic feature pathologists must evaluate during risk stratification[42]. We attribute this to their extreme rarity and the high inter-rater variability, even resulting in infrequent majority annotations among pathologists for these classes (see Figs. 3 and 4). As a result, these classes were consistently ranked as the second most likely or lower in the predictions. Nonetheless, the explanations were mostly confused within the Gleason patterns, and the overall sensitivity regarding Gleason pattern 5 is high. Aside for the rarest classes, the remaining minority classes were predicted with high accuracy, often more frequently and with greater probability mass than they were annotated (see Fig. 6a). Thus, the probability of a missed Gleason 5 pattern and a subsequent underdiagnosis is low. This behavior may be attributed to our class-averaged loss function, which emphasizes performance for minority classes or due to smaller, unannotated structures in the image, that were nonetheless predicted by GleasonXAI. Future studies could build upon this work by performing a targeted data collection for these less frequent explanations to further enhance the clinical utility of the model.

Whereas the pathologists achieved agreement similar to the literature in Gleason pattern annotation and some of the explanations[34,35,43], our analyses also revealed a higher-than-expected level of disagreement in others, such as *single cells* and *comedonecrosis*[33]. This could be caused by the overall rarity of these patterns in the dataset, but also by the low number of observers. While each image was annotated by three expert pathologists – thereby exceeding the standard of care in terms of the number of observers – increasing the number of pathologists per image could therefore further improve the estimation of the underlying diagnostic distribution for each location. This would not only reduce sampling noise in the annotations, thereby improving the learning signal for hard label approaches, but also provide more precise estimates of diagnostic uncertainty. Such improvements would yield better and more continuous targets for the soft label approaches and allow for finer evaluation of the corresponding metrics.

Our work revealed a blind-spot in segmentation research using soft-labels. While recent segmentation loss-functions have been developed for training with soft labels[36], the evaluation and the presentation of results for pathologists still hinge on hard labels. Even with a perfectly estimated diagnostic distribution, a learned minority opinion within the diagnostic distribution would not be reflected in metrics based on hard labels or in visualizations that present only the most likely explanation of the predictive distribution. Since the goal of the study was to develop a model that closely matches the pathologists' consensus, addressing this challenge is beyond the scope for this paper. However, future work on the use of predictive distributions and soft labels in medical segmentation tasks is crucial. A potential approach could involve threshold-based multi-label approaches or adapting conformal prediction techniques[44] to segmentation tasks.

The primary goal of our study was not to achieve state-of-the-art performance in Gleason pattern segmentation or grading, but rather to introduce an approach for inherent and reliable explainability in Gleason pattern segmentation with annotations using pathologists' terminology. For this, we focused our attention on the segmentation of TMAs, which, due to their smaller physical size are easier and more reliable to annotate by our pathologists. We envision the extension of our approach to WSIs to be straightforward by combining commonly used patchwise prediction pipelines with explanation-level annotated data. Promising first results were achieved with a direct application of GleasonXAI on Gleason pattern-annotated WSIs, which can be found in the Supplementary Notes: Additional Results, but it demonstrated the need for further refinement of the AI, especially in the precision of Gleason pattern 5. Future work could focus on collecting larger datasets for training, including WSIs. Our approach could likely benefit from modern techniques like semi-supervised pre-training, even larger datasets, or more advanced training techniques like additional augmentations or ensembling. We encourage other researchers to further explore and improve upon this challenging segmentation task.

In summary, we developed GleasonXAI, a pathologist-like model trained on expert-derived annotations from an international panel of pathologists, addressing the need for transparent AI in prostate cancer grading. This approach has been shown to increase users' confidence and trust in AI[45]. More importantly, it can also decrease reading time, especially for non-experts[46]. In times of declining specialist numbers and rising cancer incidence, such explainable AI systems are essential to meet clinical demand. To support further research, we publicly release the largest dataset of localized Gleason pattern explanations, aiming to advance explainable AI for high-variability medical tasks.

## Methods

### Inclusion and Ethics

The research complies with all ethics regulations. The study's ethics vote was approved by the ethics committee of the University Clinic Mannheim of the Medical Faculty of the University of Heidelberg, since our research involved no patients and no patient data was collected. Informed consent was obtained from all participating pathologists who performed the annotations of the publicly available data sets. Sex and gender of the participating pathologists were not collected or analyzed, as these variables were not relevant for the study objectives. As compensation for the annotators, we offered the opportunity to be credited as a co-author of our work.

### Development of an explanatory ontology

To gather meaningful explanations for the dataset, we developed a comprehensive medical ontology with detailed explanations for the Gleason patterns 3, 4, and 5, based on the histological description of Iczkowski KA. for Gleason grading[47]. In collaboration with an experienced uro-pathologist (NTG), these explanations were shortened, split into distinct classes and translated into German for our German collaborators (see Supplementary Tables 1–3). Our main goal was to establish a criteria-based ontology for each Gleason pattern, incorporating distinct characteristics unique to each specific pattern, that should be later annotated by experienced pathologists.

Additionally, we conducted a panel discussion with expert uro-pathologists ($n = 9$) to gather feedback on our approach. A key outcome was the need to adapt our ontology to current ISUP/WHO terminology. The wording changes were made while preserving the original content, resulting in our adapted ontology (see Fig. 8).

For our model development, we defined three distinct levels: Each Gleason pattern (depicted in dark blue) was assigned a set of broader, medically coherent explanations (depicted in light blue), which themselves contained multiple sub-explanations (depicted in white). These sub-explanations represent the original annotations that were gathered.

### Utilized datasets

In our work we focused on TMAs, as their reduced physical size allowed for more and more precisely annotated images per pathologist. We utilized data from three different data sources: 595 TMA core images were received from TissueArray.com LLC[48], 641 from Arvaniti et al. Harvard Dataverse[49,50], and 331 from the Gleason19 Challenge[13,14]. The datasets were filtered to match our requirements of containing prostate adenocarcinoma tissue with Gleason Patterns 3, 4, and 5. Initially, out of these 1567 images, 1180 TMA cores of prostate adenocarcinomas were identified as eligible for annotation with detailed explanatory features, of which 1015 images were selected for the development and evaluation of our models.

### Annotating Procedure

To ensure high-quality annotations, we recruited an international team of 54 pathologists from university clinics, non-university public clinics, and private pathology practices through the ISUP platform or direct email invitations. Of the 54 annotators, 53 took part in the annotation process, which involved two tasks: First, three to four pathologists annotated the most prevalent and malignant Gleason pattern in the TMA core images; second, three annotators applied the explanatory ontology using the sub-explanations to generate detailed and explainable annotations. To minimize potential bias related to individual pathologists' experience, we ensured that the average experience within each group of three pathologists annotating the same dataset was at least 10 years. Pathologists with less than or equal to 5 years of experience were systematically grouped with two highly experienced colleagues (≥15 years), except in one group where the average experience fell slightly below 10 years due to the unforeseen dropout of a senior pathologist without a suitable replacement.

The first task was only conducted on the TissueArray.com dataset, as the Gleason 19 challenge and Harvard Dataverse datasets already contained annotations of the Gleason patterns. The annotators were informed of the Gleason grading from the metadata of the Harvard Dataverse dataset (generated by a single pathologist based on hematoxylin and eosin staining and Immunohistochemical tests[48]), but could specify alternative patterns if they disagreed with the provided grading. The Gleason grade annotations from the pathologists were then merged using the STAPLE algorithm[51]. Similarly, the provided annotations for the Gleason19 Challenge dataset (generated by up to six pathologists[14]) were merged using the same algorithm. The resulting output masks were reviewed for quality and filtered by an observer (SLP).

After the merging of the Gleason Grade annotation masks, additional filtering was required. Of the 1567 images, 244 were removed since they didn't fit the tissue requirements of containing prostate carcinoma with a Gleason Grade above 2. An additional 143 were removed due to missing annotators, small or no Gleason grade areas, containing only Gleason Grade 1 or 2 annotations post-merge, or other quality concerns. In 19 cases in the Gleason 19 Challenge dataset, the merged grade annotations produced by STAPLE did not align with any meaningful biological patterns (e.g., too small or fragmented areas), though individual annotator labels did. For these cases, the annotation of the pathologist, whose annotation was deemed the closest match to the STAPLE output by an observer (SLP), was selected (see Supplementary Fig. 9 for a representative image).

For the second task, the TMA core images and their corresponding Gleason pattern annotation maps from the first task were divided in 15 distinct sets. Each set was provided to a group of annotators, who were then tasked with annotating specific histological patterns within the predefined areas to explain the respective Gleason pattern and assign a corresponding explanatory text. A free-text option was available if the pathologists disagreed with the provided explanation choices. For each annotated image from the first task, the pathologists of the second task received up to two different images, each with the outline of the annotation area for a single Gleason pattern (single-grade images). In cases

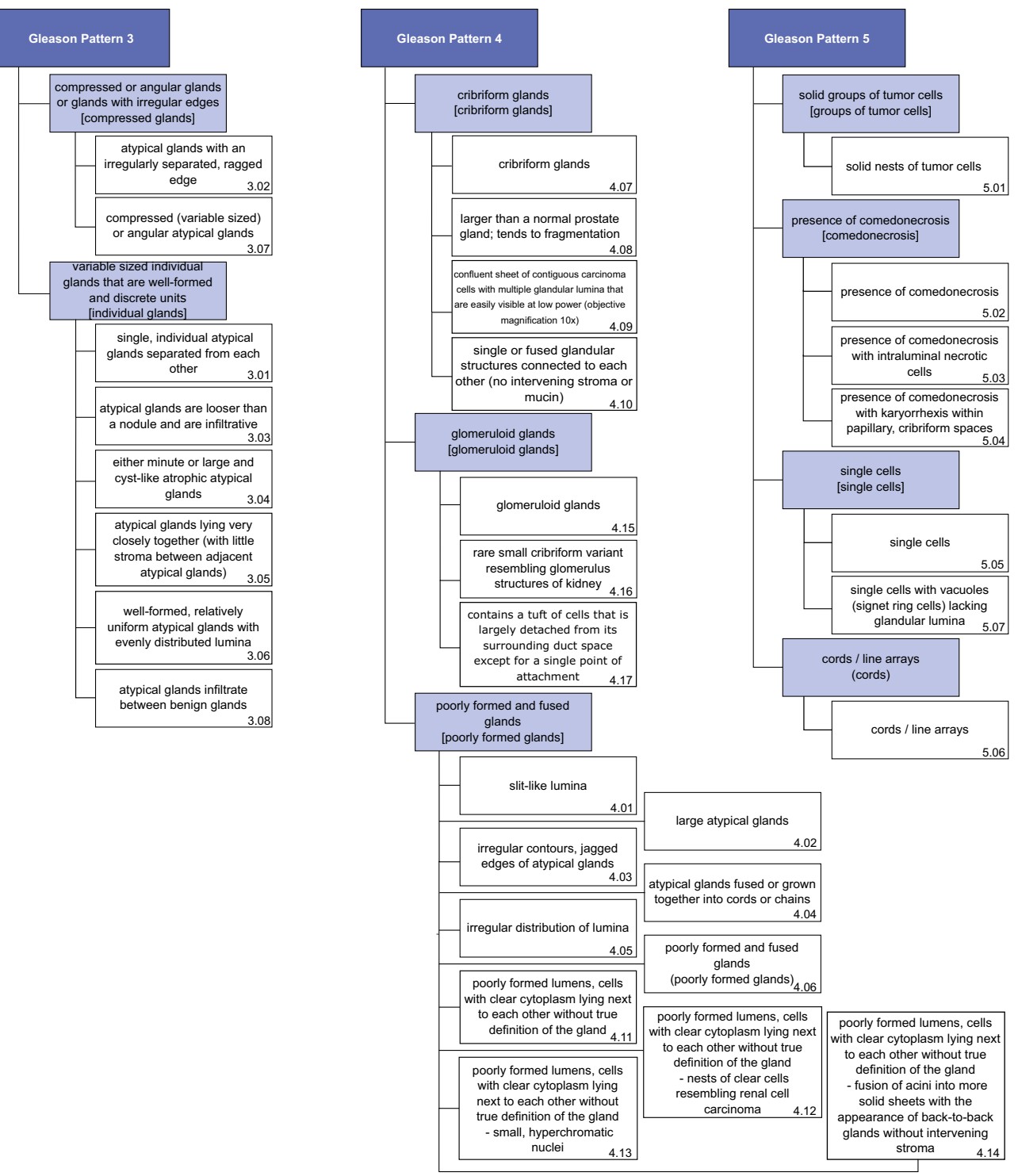

**Fig. 8 | Overview of the explanatory Gleason pattern ontology.** Generic terms based on the WHO and ISUP2014 guidelines summarize the explanations corresponding to our initial ontology version. As the term "hypernephroid pattern" is now discouraged, we replaced it with "poorly formed lumens, cells with clear cytoplasm lying next to each other without true definition of the gland". Gleason pattern classes are marked in dark blue, explanation classes in light blue, and sub-explanations in white. For our figures, we use shortened names of the explanations, which are shown in square brackets for explanations, and in the numbering in the bottom right for the sub-explanations.

where the assigned Gleason grades were identical (i.e., 3 + 3, 4 + 4, or 5 + 5), only one single image with the corresponding area marked was presented (see Supplementary Fig. 10).

After the annotation with the explanatory labels, additional images ($n = 165$) had to be excluded due to an insufficient number of raters. The objective was to obtain annotations from three

pathologists for each image. We used the annotations of the first three pathologists who responded and completed their tasks. Pathologists who dropped out early in the annotation process were replaced, and their contributions were not included in the final dataset, if they annotated less than a quarter of their assigned data. The images for which we did not receive full annotations from three pathologists by

the end of the process were removed. This occurred primarily due to late dropouts, but also due to single images being skipped. We reviewed the skipped images to account for potential systematic biases, but were unable to identify any consistent issues. Overall, annotations for 1015 TMA core images from 42 of the 47 involved annotators in the explanation annotation phase were included.

All annotations were performed using the online annotation tool PlainSight[52]. Further details on the data selection is available in the Supplementary Notes: Data Selection.

## Data preparation

After the raters completed the explanatory annotations, explanations provided via the free text option had to be standardized. This was achieved by mapping them to their nearest equivalent in the ontology.

Sorted by the time of polygon creation, empty explanation fields were filled with the next available explanation. This reflected the expected behavior outlined in the instruction video provided to the raters: polygons were drawn first, and afterwards the explanations were selected. If multiple explanations were selected for a polygon, each explanation was included in the data as a copy of the original polygon.

For our model training, we created segmentation masks for each annotator and TMA core image pairs. As the explanations for each Gleason pattern were annotated separately on single-grade images, we drew the annotations of each single-grade image in the order they were created, with the single-grade images themselves being sorted by their corresponding Gleason pattern in ascending order. Annotations that were provided later therefore override previous annotations of the same or lower Gleason patterns that share the same pixels.

The tree-structure of the ontology allowed us to create three datasets from our annotations (sub-explanations, explanations, and Gleason patterns, see Fig. 8), by remapping the sub-explanation annotations upwards in the ontology. This was deemed necessary for the model development, due to concerns of the sample size for the sub-explanations and the issue of class imbalance.

For both training and label analysis, we used merged grade images to review the complete grading and interpretations provided by the pathologists for the TMA core images.

## Model Development

To develop a pathologist-like, inherently explainable AI system (Gleason XAI) for detecting Gleason patterns on TMA cores, we employed a concept bottleneck strategy[29], predicting the explanations directly for each pixel, with the ability to later remap them to their corresponding Gleason pattern. This provides inherent explainability by basing the decision for a Gleason pattern solely on the predictions of the associated explanations, which can in turn be verified by expert pathologists in contrast to black box decisions on the Gleason pattern. Inspired by recent positive results[53], we selected a U-Net[30] with an Image-Net[54]-pretrained EfficientNet-B4[55] encoder as our segmentation architecture. To validate our architectural choice, we compared the results on additional different architectures. The comparison can be found in the Supplementary Notes: Additional Results.

We trained models on all three levels of our ontology (see Fig. 8). At each level of the class hierarchy, an additional *benign tissue* class was included to account for unannotated regions.

As the white background of the slide would be falsely labeled as *benign tissue*, we used Otsu's thresholding[56] followed by morphological closing and opening operations to detect and mask out these regions during the loss-computation (see Supplementary Fig. 11a). During inference, the entire image was processed, however, the loss functions and the metrics were computed using only foreground pixels. For our segmentation visualizations, we also masked out the background pixels.

Due to high interobserver variability in the annotations, particularly for the explanations and sub-explanations, using majority voting or more sophisticated annotation merging approaches like STAPLE to

obtain definite per-pixel labels was not feasible. Instead, we used a soft label approach, obtained by combining the annotations into a per-pixel annotation distribution (see Supplementary Fig. 11b). This method preserved human uncertainty in the labeling process and retained annotations for minority classes or cases where multiple explanations might be feasible.

When training segmentation models with majority-voted labels, often a combination of cross-entropy loss and variants of the Dice loss[57] is used. However, the Dice loss does not apply to soft labels. We therefore implemented the SoftDiceLoss, a straightforward extension of the Macro Dice loss, only changing the domain of the target variables to the interval [0,1]. This is similar to the one proposed by Wang et al.[36], however, we found that their loss function led to training instabilities in our training, providing worse results in terms of our metrics between different seedings.

For comparison with existing literature, we also nonetheless included models trained with the cross-entropy and class-averaged Dice loss on majority-voted labels. For better comparability between the soft- and hard-label methods, we decided against using STAPLE to preserve the decision boundaries. These loss-functions were only computed over foreground pixels that possess an unambiguous majority vote (84.41% of the pixels on explanation level, 97.54% on Gleason patterns, see Fig. 4). Of the pixels where all three pathologists disagreed on the explanation, 64.82% were labeled as benign tissue by one annotator and most of the time as different explanations of the same Gleason pattern by the other two. They can therefore be assumed to be partially caused by differences in the delineation, and thus should have only a minor effect if removed.

Each model was trained three times with different seeds, using a batch size of 12 for 200 epochs. The starting learning rate was set to 5e-5, and it was reduced by a factor of three if the validation loss did not decrease for two consecutive epochs. The $L_2$ parameter regularization coefficient was set to $\lambda = 0.02$. Both parameters were found through hyperparameter optimization using the optuna[58] library. We used AdamW[59] as optimizer with default parameters ($\beta_1 = 0.99$, $\beta_2 = 0.9$). Models were saved after each epoch, and the epoch with the lowest validation loss was selected for testing.

The TMA core images possessed resolutions between 2232 × 2215 px² and 5632 × 5632 px², stemming from different scanners and different physical sizes of the images. As the individual datasets had different pixel spacings (Gleason19 Challenge: 0.25 $\frac{\mu m}{px}$, Harvard Dataverse: 0.23 $\frac{\mu m}{px}$, TissueArray.com: 0.5455 $\frac{\mu m}{px}$), we tested multiple resolutions for segmentation performance and then bi-cubically interpolated all images to a common physical pixel side length of 1.392 $\frac{\mu m}{px}$. At this resolution, the smallest images filled a 512 × 512 px² patch and the largest images reached 1358 × 1358 px². We augmented the images during training using the light augmentations without stain-normalizations, as recommended by Tellez et al.[60] and extracted random patches of size 512 × 512 px², while avoiding patches consisting only of background. For the validation set, we always extracted the central 512 × 512 px² patch of each image. At test time, we utilized the computationally more expensive sliding window approach, implemented by MONAI[61], extracting 512 × 512 px² patches, with 50% overlap and Gaussian weighted averaging with default parameters. A graphical overview of our approach can be found in Supplementary Fig. 11.

The dataset was randomly split into training, validation and test datasets, comprising 70%, 15%, and 15% of the TMA cores, respectively. We optimized the assignment of images to the training, validation, and test splits to minimize the L1-norm (see Model Evaluation) between the class distributions of the pixels between the splits. The class distributions are shown in Supplementary Fig. 12.

## Statistics and software

**Software.** All code was written in Python (3.10.13). PyTorch (2.1.1), PyTorch Lightning (2.2.0.post0), Albumentations (1.3.1), Pillow (9.5.0),

Openslide (1.4.2), Pyvips (3.0.0), Shapely (2.1.1), OpenCV (4.8.1.78), MONAI (1.3.0), Hydra (1.3.2), NumPy (2.2.6), Pandas (2.1.1), Timm (0.9.2), WandB (0.17.1), Tensorboard (2.16.2), Omegaconf (2.3.0), SciPy (1.11.3), Scikit-learn (1.3.2), Scikit-image (0.22.0), Statsmodels (0.13.2), Matplotlib (3.8.0), and Seaborn (0.13.2) were used for image processing, model development and training, data analysis, and visualization.

**Pathologists' caseload.** The participating pathologists were asked to provide an estimate of the number of prostate cancer patients they examined per week, with their responses provided in ranges. To address the issue of overlapping ranges, a representative value was determined for each pathologist by calculating the mean of their submitted range. The median of these means was reported in the results.

**Interobserver agreement.** To analyze the annotator concordance while accounting for agreement occurring by chance, we calculated Fleiss' kappa[31] within each rater group and label using the statsmodels[62] implementation.

For this calculation, we binarized the pathologists' decisions by evaluating, for each image and label, whether the label was used by the annotator, resulting in nine decisions per image and annotator. Fleiss' kappa values were then calculated for each label within each group. To assess overall agreement within each group, we calculated Fleiss' kappa across all decisions of the group regardless of the labels. Additionally, we quantified the overall agreement for each label by calculating Fleiss' kappa across all decisions for that label in all TMA core images.

**Model evaluation.** In our setting, which used soft labels for segmentation tasks, traditional metrics like Dice score, Intersection-over-Union, accuracy, and ROC curves offer an incomplete picture, as they rely on hard labels and do not take into account the reviewers' clinical uncertainty.

Therefore, we reported the complement of the mIoU$^D$ ($\alpha = 0.5$, $\beta = 0.5$) loss by Wang et al.[36], a segmentation Dice loss that was developed for soft labels, as our primary metric (referred to as Macro SoftDice for brevity). Importantly, this loss averages over all classes, which is important for datasets containing large class imbalances. As discussed in Methods: Model Development, using this loss during training leads to worse performance for our model in terms of all metrics compared to our SoftDiceLoss. Due to its sound theoretical motivation and to avoid having to use our own loss function as target metric, we used the Macro SoftDice.

Additionally, we focused on evaluating the predictive calibration by measuring the divergence between the predictive and target distributions. Contrary to hard label-based metrics, this approach accounted for pathologists' diagnostic uncertainty. We reported the $L_1$-norm normalized to the range (0,1), from now on called $L_1$-norm, between the predictive and target distributions:

$$L_1(p^{seg}, y^{seg}) = \frac{1}{2NCP} \sum_{n,c,p=1}^{N,C,P} \left| p_{n,c,p} - y_{n,c,p} \right| \tag{1}$$

A pathologist will most likely interact with the visualization of the most probable class per pixel, as model explanations need to be simple and efficient in clinical practice[16]. Therefore, to maintain compatibility with adjacent literature and to provide intuitively understandable metrics, we also reported the Dice and the Macro Dice scores, computed globally over all pixels. These were calculated between the pixel-wise maximum of our predictive distribution and the majority vote of the annotation distribution as our secondary metrics. We also included a confusion matrix of the explanation predictions.

In our dataset, a unique majority vote could not be defined for pixels, where all three pathologists disagreed. Therefore, we computed these metrics only over foreground pixels with an unambiguous majority vote. The primary goal of this paper was to develop a pathologist-like, inherently explainable segmentation model for Gleason patterns on TMAs. Accordingly, we also evaluated the performance of the models in Gleason pattern segmentation. We compared models trained on the (sub-)explanations of our ontology to models trained solely on the Gleason patterns. For our models trained on the (sub-)explanations, we remapped the predictions to the Gleason patterns by aggregating the corresponding probabilities of the predictions, summing up the predicted probabilities of the (sub-)explanations that − according to our predefined ontology − corresponded to a Gleason pattern.

We reported the mean and standard deviation of the metrics, averaged over three runs.

## Reporting summary
Further information on research design is available in the Nature Portfolio Reporting Summary linked to this article.

## Data availability
The annotation data based on all three datasets generated in this study and the TMA core images of the TissueArray.com LLC dataset used in this study have been deposited in the Figshare repository at https://doi.org/10.6084/m9.figshare.27301845 [63]. The TMA core images of the Gleason 19 challenge used in this study are available on the Grand Challenge platform at https://gleason2019.grand-challenge.org/Register/. The TMA core images of the Arvaniti et al. The Harvard Dataverse dataset used in this study is available in the Harvard Dataverse repository at https://doi.org/10.7910/DVN/OCYCMP. The WSI images of the AGGC dataset used in this study in the Supplementary Notes are available on the Grand Challenge platform. Access can be obtained after registration at https://aggc22.grand-challenge.org. The WSI images of the DiagSet dataset used in this study in the Supplementary Notes can be obtained by registering at the database and providing a description of the intended use at https://ai-econsilio.diag.pl. Source data are provided with this paper.

## Code availability
The code for the model development and the statistical analyses is available on GitHub at https://github.com/DBO-DKFZ/GleasonXAI.

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

## Acknowledgements

This project was funded by the Ministry of Health, Integration and Social Affairs, Baden-Württemberg, Germany (project name: "KI-Translations-Initiative"). The funder had no role in the design, data collection, data analysis, and reporting of this study. JNK is supported by the German Cancer Aid DKH (DECADE, 70115166), the German Federal Ministry of Research, Technology and Space BMFTR (PEARL, 01KD2104C; CAMINO, 01EO2101; TRANSFORM LIVER, 031L0312A; TANGERINE, 01KT2302 through ERA-NET Transcan; Come2Data, 16DKZ2044A; DEEP-HCC, 031L0315A; DECIPHER-M, 01KD2420A; NextBIG, 01ZU2402A), the German Research Foundation DFG (CRC/TR 412, 535081457; SFB 1709/1 2025, 533056198), the German Academic Exchange Service DAAD (SECAI, 57616814), the German Federal Joint Committee G-BA (TransplantKI, 01VSF21048), the European Union EU's Horizon Europe research and innovation program (ODELIA, 101057091; GENIAL, 101096312), the European Research Council ERC (NADIR, 101114631), the National Institutes of Health NIH (EPICO, R01 CA263318) and the National Institute for Health and Care Research NIHR (Leeds Biomedical Research Centre, NIHR203331). The views expressed are those of the author(s) and not necessarily those of the NHS, the NIHR or the Department of Health and Social Care. This work was funded by the European Union. Views and opinions expressed are however, those of the author(s) only and do not necessarily reflect those of the European Union. Neither the European Union nor the granting authority can be held responsible for them.

## Author contributions

SLP drafted the initial version of the manuscript. SLP conducted the recruitment and data collection phase. H.A.M., G.M., and S.H. wrote the version sent out to the other authors of the manuscript. SH and TCB were involved in the supervision of the project. S.L.P., H.A.M., T.C.B., G.M., S.H. and T.J.B. conceived the project. H.A.M. and G.M. developed all methods and conducted the preparation and analysis of the data and results. TJB supervised the project from initiation to submission and takes responsibility for all methods developed, the published data as well as all analyses conducted. TJB acquired the funding for this project. M.W., G.G.K., T.T.R., C.N., E.M.C., A.G., M.H., N.J.R., J.W., I.K., M.S., C.M.S., M.Ba., W.S., Y.C.T., F.W., R.G., J.A., T.K., C.H., K.D.M., C.D., A.E., G.D., F.M., U.S., M.Br., J.G., M.S.L.L., K.T., L.C., F.C., A.L., G.C., T.QN., A.A., A.C., A.S., V.M., N.A., N.M., F.Sa., D.T., E.B., L.J., R.B.S., F.Si., NS, KP, OH, SS, MRD, HM, SRW annotated data for the study. SH, TCB, TC, NTG, MW, GGK, TTR, CN, EMC, A.G., M.H., N.J.R., J.W., I.K., M.S., C.M.S., M.Ba., W.S., Y.C.T., F.W., R.G., J.A., T.K., C.H., K.D.M., C.D., A.E., G.D., F.M., U.S., M.Br., J.G., M.S.L.L., K.T., L.C., F.C., A.L., G.C., T.Q.N., A.A., A.C., A.S., V.M., N.A., N.M., F.Sa., D.T., E.B., L.J., R.B.S., F.Si., N.S., K.P., O.H., S.S., M.R.D., H.M., S.R.W., C.V.S., J.N.K., Y.T., and T.J.B. provided clinical expertise and/ or machine learning expertise, contributed to the interpretation of the results and critically revised the manuscript. T.H.L. and C.W. contributed to the study planning, statistical evaluation, and interpretation of the results, and critically revised the manuscript.

## Funding

## Competing interests

TJB owns a company that develops mobile apps (Smart Health Heidelberg GmbH, Heidelberg, Germany). TJB received honoraria from Novartis, Roche and HEINE Optotechnik. JNK declares consulting services for Panakeia, AstraZeneca, MultiplexDx, Mindpeak, Owkin, DoMore Diagnostics, and Bioptimus. Furthermore, he holds shares in StratifAI, Synagen, Tremont AI, and Ignition Labs, has received an institutional research grant from GSK, and has received honoraria from AstraZeneca, Bayer, Daiichi Sankyo, Eisai, Janssen, Merck, MSD, BMS, Roche, Pfizer, and Fresenius. YT declares consulting services for Indica Labs, AstraZeneca, MSD, Pfizer; royalties not related to this study (Indica Labs). CMS is a cofounder and shareholder of Vicinity Bio GmbH, and is a scientific advisor to and has received research funding from Enable Medicine Inc., all outside the current work. HM declares consulting services for PathXL, Invicro, and PathAI, outside of this work. NJR discloses an advisory board function for AbbVie AG, and receipt of a travel grant from Roche Diagnostics, both outside of the scope of the current work. IK has received honoraria from AstraZeneca and Menarini Stemline, as well as gifts/ financial advantages from Roche (conference invitations). NTG received an institutional research grant from Janssen/Johnson & Johnson and declares consulting services for/honoraria from AstraZeneca, Janssen, Merck, BMS, Daiichi Sankyo, and Bayer. No other conflicts of interest are declared by any of the authors.

## Additional information

Gesa Mittmann [1,2,66], Sara Laiouar-Pedari[1,66], Hendrik A. Mehrtens[1,66], Sarah Haggenmüller[1], Tabea-Clara Bucher[1], Tirtha Chanda [1,2], Nadine T. Gaisa[3,4], Mathias Wagner[5], Gilbert Georg Klamminger[5], Tilman T. Rau[6], Christina Neppl[6], Eva Maria Compérat[7], Andreas Gocht[8], Monika Haemmerle [9], Niels J. Rupp[10,11], Jula Westhoff[12], Irene Krücken[13,14], Maximilian Seidl [6], Christian M. Schürch [15,16], Marcus Bauer [9], Wiebke Solass[17], Yu Chun Tam[18], Florian Weber [19], Rainer Grobholz[11,20], Jaroslaw Augustyniak[21], Thomas Kalinski[22], Christian Hörner[23], Kirsten D. Mertz[24,25], Constanze Döring[26], Andreas Erbersdobler[27], Gabriele Deubler[28], Felix Bremmer[29], Ulrich Sommer[30], Michael Brodhun[31], Jon Griffin [32], Maria Sarah L. Lenon[33,34], Kiril Trpkov[35], Liang Cheng [36], Fei Chen [37], Angelique Levi[38], Guoping Cai[38], Tri Q. Nguyen[39], Ali Amin[40], Alessia Cimadamore[41], Ahmed Shabaik[42], Varsha Manucha[43], Nazeel Ahmad[44], Nidia Messias[45], Francesca Sanguedolce[46], Diana Taheri[47,48], Ezra Baraban[49], Liwei Jia[50], Rajal B. Shah[50], Farshid Siadat[35], Nicole Swarbrick[51,52], Kyung Park[37], Oudai Hassan[53], Siamak Sakhaie[54], Michelle R. Downes[55], Hiroshi Miyamoto [56], Sean R. Williamson[57], Tim Holland-Letz[58], Christoph Wies [1,2], Carolin V. Schneider[59], Jakob Nikolas Kather[60,61,62,63], Yuri Tolkach [64,65] & Titus J. Brinker [1] ✉

[1]Division of Digital Prevention, Diagnostics and Therapy Guidance, German Cancer Research Center (DKFZ), Heidelberg, Germany. [2] Medical Faculty, Heidelberg University, Heidelberg, Germany. [3]Institute of Pathology, RWTH Aachen University, Aachen, Germany. [4]Institute of Pathology, University Hospital, University of Ulm, Ulm, Germany. [5]Department of Pathology, University of Saarland, Homburg Saar Campus, Homburg Saar, Germany. [6]Heinrich-Heine-University and University Hospital Düsseldorf, Düsseldorf, Germany. [7]Department of Pathology, Medical University of Vienna, Vienna, Austria. [8]Institute of Pathology, University Hospital Schleswig-Holstein, Lübeck, Germany. [9]Institute of Pathology, Martin Luther University Halle-Wittenberg, Halle (Saale), Germany. [10]Department of Pathology and Molecular Pathology, University Hospital Zurich, Zurich, Switzerland. [11]Faculty of Medicine, University of Zurich, Zurich, Switzerland. [12]Institute of Pathology, Städtisches Klinikum Karlsruhe, Karlsruhe, Germany. [13]Institute of Pathology, Klinikum Bremen Mitte, Bremen, Germany. [14]PathoNext GmbH, Leipzig, Germany. [15]Department of Pathology and Neuropathology, University Hospital and Comprehensive Cancer Center Tübingen, Tübingen, Germany. [16]Cluster of Excellence iFIT (EXC 2180) "Image-Guided and Functionally Instructed Tumor Therapies", University of Tübingen, Tübingen, Germany. [17]Institute of Tissue Medicine and Pathology, University Bern, Bern, Switzerland. [18]Institute of Pathology, Georgius Agricola Foundation Ruhr, Ruhr University Bochum, Bochum, Germany. [19]Institute of Pathology, University of Regensburg, Regensburg, Germany. [20]Institute of Pathology, Cantonal Hospital Aarau, Aarau, Switzerland. [21]Laborteam Pathology, Goldach, Switzerland. [22]Institute of Pathology, Brandenburg Medical School Theodor Fontane, University Hospital Brandenburg an der Havel, Brandenburg an der Havel, Germany. [23]Institute for Pathology, University Medical Faculty Mannheim, University of Heidelberg, Mannheim, Germany. [24]Institute of Medical Genetics and Pathology, University Hospital Basel, Basel, Switzerland. [25]Department of Biomedicine (DBM), Pathology of Infectious and Immunologic Diseases, University of Basel, Basel, Switzerland. [26]Pathology Zwickau, MVZ diagnosticum Stollberg, Zweigpraxis Zwickau, Zwickau, Germany. [27]Institute of Pathology, University Medicine Rostock, Rostock, Germany. [28]Institute of Pathology, Kreiskliniken Reutlingen, Reutlingen, Germany. [29]Institute of Pathology, University Medical Center Goettingen, Goettingen, Germany. [30]Institute of Pathology, University Hospital of Dresden, Dresden, Germany. [31]Institute of Pathology and Neuropathology, HELIOS Klinikum Erfurt, Erfurt, Germany. [32]School of Medicine and Population Health, University of Sheffield, Sheffield, UK. [33]Department of Pathology, University of Santo Tomas Hospital, Manila, Philippines. [34]Department of Pathology and Laboratory Medicine, National Kidney and Transplant Institute, Quezon City, Philippines. [35]Department of Pathology and Laboratory Medicine, Cumming School of Medicine, University of Calgary, Rockyview General Hospital, Calgary, AB, Canada. [36]Department of Pathology and Laboratory Medicine, Department of Surgery (Urology), Brown University Warren Alpert Medical School, the Legorreta Cancer Center at Brown University, and Brown University Health, Providence, RI, USA. [37]Department of Pathology, NYU Langone Health, New York, NY, USA. [38]Department of Pathology, Yale University School of Medicine, New Haven, CT, USA. [39]Department of Pathology, University Medical Centre Utrecht, Utrecht, The Netherlands. [40]Department of Pathology, Warren Alpert Medical School of Brown University, Providence, RI, USA. [41]Pathological Anatomy, University of Udine, Udine, Friuli-Venezia Giulia, Italy. [42]Department of Pathology, UC San Diego School of Medicine, La Jolla, CA, USA. [43]Department of Pathology, University of Mississippi Medical Center, Jackson, MS, USA. [44]University of South Florida, Morsani College of Medicine, Department of Pathology & Cell Biology and James A. Haley Veterans' Hospital, Tampa, FL, USA. [45]Department of Pathology and Immunology, Washington University in St. Louis, St. Louis, MO, USA. [46]Department of Pathology, University of Foggia, Foggia, Italy. [47]Isfahan Kidney Diseases Research Center, Department of Pathology, Isfahan University of Medical Sciences, Isfahan, Iran. [48]Urology Research Center, Tehran University of Medical Sciences, Tehran, Iran. [49]Department of Pathology, Johns Hopkins

University, Baltimore, MD, USA. [50]Department of Pathology, University of Texas Southwestern Medical Center, Dallas, TX, USA. [51]Department of Anatomical Pathology, PathWest Laboratory Medicine WA, Perth, WA, Australia. [52]Division of Pathology and Laboratory Medicine, UWA Medical School, Crawley, WA, Australia. [53]Department of Pathology, Henry Ford Health System, Detroit, MI, USA. [54]Department of Anatomical Pathology, Frankston Laboratory, Dorevitch Pathology, Frankston, VIC, Australia. [55]Anatomic Pathology, Precision Diagnostics & Therapeutics Program, Sunnybrook Health Sciences Centre, Toronto, ON, Canada. [56]Department of Pathology and Laboratory Medicine, University of Rochester Medical Center, Rochester, NY, USA. [57]Department of Pathology, Diagnostics Institute, Cleveland Clinic, Cleveland, OH, USA. [58]Division of Biostatistics, German Cancer Research Center(DKFZ), Heidelberg, Germany. [59]Department of Internal Medicine, University Hospital Aachen, RWTH University of Aachen, Aachen, Germany. [60]Else Kroener Fresenius Center for Digital Health, Faculty of Medicine and University Hospital Carl Gustav Carus, TUD Dresden University of Technology, Dresden, Germany. [61]Department of Medicine I, Faculty of Medicine and University Hospital Carl Gustav Carus, TUD Dresden University of Technology, Dresden, Germany. [62]Medical Oncology, National Center for Tumor Diseases (NCT), University Hospital Heidelberg, Heidelberg, Germany. [63]Pathology & Data Analytics, Leeds Institute of Medical Research at St James's, University of Leeds, Leeds, UK. [64]Institute of Pathology, University Hospital Cologne, Cologne, Germany. [65]Medical Faculty, University of Cologne, Cologne, Germany. [66]These authors contributed equally: Gesa Mittmann, Sara Laiouar-Pedari, Hendrik A. Mehrtens. ✉e-mail: titus.brinker@dkfz.de

