## [Transparent Peer Review file · Nature Communications]

Pathologist-like explainable AI for interpretable Gleason grading in prostate cancer

Corresponding Author: Dr Titus Brinker

Version 0:

Reviewer comments:

Reviewer #1

(Remarks to the Author)

Mittmann et al. have developed the GleasonXAI system, an explainable AI that uses U-Net for interpretable Gleason grading. This system leverages a novel dataset of 1,015 images annotated by 54 pathologists, following GUPS/ISUP guidelines. However, it utilizes tissue microarrays (TMAs) instead of whole slide images (WSIs) from core needle biopsies (CNBs) or whole excision biopsies. Despite this limitation, the system employs soft labels to address interobserver variability, achieving a competitive Dice score of 0.713, which enhances transparency in AI-based diagnostics.

Compared to the earlier submission to Nature Cancer, the authors have addressed several prior concerns, including clarifying figures, expanding the discussion of limitations, providing code availability, and presenting a more balanced interpretation of the results. However, several important issues remain:

- The lack of comparison between U-Net and other explainable segmentation architectures (e.g., FCN, SegNet) remains a limitation. The intention of the original comment was to encourage benchmarking across interpretable models, rather than general CNNs. If additional comparisons are not feasible, this limitation should be clearly acknowledged in the discussion.
- A large number of variables and annotations were included in the study in an attempt to improve model performance and explainability (including the profile of three participants selected for each core, demographic profile of the cases included, 32 ontology terminologies, three Gleason patterns, and the presence of technical replicates for each core). While the authors emphasized the accuracy of the annotations by constraining the diversity and complexity of the dataset to just TMA cores and a sample size of 1,015, a formal sample size calculation should be considered to determine the minimum number of cases required to ensure accuracy and reliability. By virtue of the study design, generalizability to clinical practice remains a concern unless a broader cohort encompassing multi-site biopsies or whole slide images is included. At least for validation purposes, an additional cohort comprising these entities could be used to assess performance in predicting only the Gleason score (against the clinical Gleason score available from sign-outs), considering that annotating large slides would be challenging.
- There are a total of seven ontologies in Fig. 4a that were identified by only one pathologist. This raises questions about how the three pathologists for each core were selected, and whether there were significant differences in their professional expertise and profiles. Such variation could introduce potential bias in the annotations.
- It is uncertain whether similar results would be expected from large external datasets like TCGA and CPTAC without performing the analysis. While tailored annotations are not available, updated annotation of the existing whole slide images is feasible—at least in terms of Gleason score and pattern.

(Remarks on code availability)

Reviewer #2

(Remarks to the Author)

The authors answered all my comments.

(Remarks on code availability)

Reviewer #3

(Remarks to the Author)

Thank you for revising this manuscript. While the Authors have made a commendable effort to respond comprehensively to the reviewer's comments, several key concerns remain insufficiently addressed.

1. In response to the critique regarding software quality, the authors note improvements in documentation and usability, but they do not explicitly address the absence of unit tests or provide assurance of robust versioning practices—both critical for reproducibility and long-term maintainability.

2. Regarding novelty, the response reiterates the work's intended contributions but does not convincingly refute the reviewer's concern that similar ideas around hierarchical modeling and interpretability have been explored previously, leaving the originality of the work ambiguous.

3. Furthermore, in addressing data exclusion and pixel-level disagreements, while the Authors provide an exhaustive breakdown, they offer limited analytical insight into how these exclusions impact the generalizability or clinical utility of the model.

4. Finally, the justification for introducing the SoftDiceLoss remains largely empirical and lacks rigorous benchmarking or theoretical grounding, leaving open questions about potential overfitting.

These gaps suggest the need for a more transparent, critically reflective engagement with the reviewer's core concerns. Similar concerns were raised by other reviewers, and the degree to which these have been addressed is variable.

(Remarks on code availability)

Version 1:

Reviewer comments:

Reviewer #1

(Remarks to the Author)

The authors have adequately addressed reviewer concerns. Additional model comparisons now included in the Supplementary Materials support the validity of their findings.

(Remarks on code availability)

Reviewer #2

(Remarks to the Author)

The authors addressed all my comments.

(Remarks on code availability)

Reviewer #3

(Remarks to the Author)

Thank you for addressing the comments.

(Remarks on code availability)

We would like to thank all the reviewers for their efforts in revising our paper. We are grateful for recognising our 'commendable effort to respond comprehensively' (Reviewer #2) to the previous remarks, and we are glad that we were able to address several prior concerns (Reviewer #1) if not all of the previous comments (Reviewer #3). Below, we address the remaining concerns raised by the reviewers:

REVIEWER COMMENTS

Original Reviewer #1 (Remarks to the Author):

Comment R1.1:

The lack of comparison between U-Net and other explainable segmentation architectures (e.g., FCN, SegNet) remains a limitation. The intention of the original comment was to encourage benchmarking across interpretable models, rather than general CNNs. If additional comparisons are not feasible, this limitation should be clearly acknowledged in the discussion.

We thank the reviewer for his feedback. The model selection in our study was straightforward: we employed the widely recognized U-Net architecture, which remains the state-of-the-art for segmentation tasks in medical imaging. This decision was aligned with our primary goal of demonstrating the feasibility and value of interpretability within an established and clinically relevant framework and was based on the observation that multiple prior studies [1,2] have shown that the UNet architecture remains state-of-the-art for medical image segmentations. Based on these studies, we originally decided not to include a comparison to other architectures as our goal was not to reach a new high-score on an existing dataset, but to demonstrate the value of adapting concept bottleneck approaches in machine learning models for Gleason grading. A key advantage of our interpretability approach lies in its close link to the underlying ground truth. Unlike other interpretable methods, such as Grad-CAM-based options, our framework does not rely on indirect approximations or intermediary visualizations, which are known to be misinterpreted and are prone to confirmation bias.

To accommodate the reviewer's request, we trained a suite of additional models and compared their performance to our original U-Net-based model (with an EfficientNet-B4 encoder). As a DenseNet encoder was requested in the last revision, we additionally trained a UNet with a DenseNet-201 [3] (as they are closest in the number of parameters) encoder, again pre-trained on ImageNet, as well as UNet++ [4] architecture with an EfficientNetB4 encoder. Furthermore, we also employed a DeepLabV3+ [5] network with an EfficientNetB4 encoder and, as requested, the original FCN [6], which employed a ResNet50 encoder.

We trained all these models to segment the explanations, using the DICE loss for the majority voted hard-labels or the SoftDICELoss for the soft-label approach. Below you can find a table with the results for the previously reported metrics.

Loss function	Architecture / Encoder	Gleason				Explanations			
		Macro DICE		Macro SoftDICE		Macro DICE		Macro SoftDICE	
		mean	std	mean	std	mean	std	mean	std
SoftDICE	UNet / EfficientNetB4	0.711	0.004	0.677	0.001	0.367	0.005	0.351	0.004
SoftDICE	UNet / DenseNet-201	0.691	0.011	0.657	0.008	0.356	0.003	0.333	0.008
SoftDICE	UNet++ / EfficientNetB4	0.662	0.035	0.629	0.029	0.343	0.009	0.319	0.012
SoftDICE	FCN / ResNet50	0.680	0.007	0.651	0.008	0.366	0.010	0.343	0.007
SoftDICE	DeepLabV3+ / EfficientNetB4	0.672	0.013	0.647	0.011	0.360	0.011	0.330	0.006
DICE	UNet / EfficientNetB4	0.610	0.112	0.590	0.095	0.358	0.013	0.319	0.012
DICE	UNet / DenseNet-201	0.521	0.082	0.524	0.071	0.356	0.004	0.313	0.004
DICE	UNet++ / EfficientNetB4	0.533	0.076	0.525	0.060	0.354	0.012	0.315	0.010
DICE	FCN / ResNet50	0.517	0.033	0.527	0.028	0.346	0.003	0.311	0.001
DICE	DeepLabV3+ / EfficientNetB4	0.581	0.002	0.579	0.014	0.342	0.009	0.309	0.001

Table 1: Results with additional architectures. Models trained with the SoftDICE loss used soft-labels, while DICE-trained models used hard-labels (leaving out pixels without majority vote). All models were trained on the explanations and evaluated on the explanations and the Gleason level. Our original results from the paper are shaded in light blue. The best result per column is marked in **bold**.

As demonstrated by our results, the use of different architectural variants yields largely comparable performance. We initially selected the EfficientNet architecture due to its strong performance and high computational efficiency, outperforming alternatives such as DenseNet and ResNet in this regard ⁵⁵. As expected, the EfficientNet-based model outperformed the version using a DenseNet-201 encoder. Similarly, prior studies have shown through comprehensive benchmarking that modifications to the U-Net architecture, such as UNet++, UNet3+, and others, often achieve similar performance levels ^{68,53,68}. This observation is consistent with our findings, where both DeepLabV3+ and UNet++ failed to improve upon our fine-tuned U-Net baseline. The original Fully Convolutional Network (FCN), which lacks a dedicated decoder path and predates the U-Net architecture, exhibited inferior segmentation performance, as expected.

All experiments were conducted using the same hyperparameters, due to constraints in time and computational resources. However, based on our results and the literature, we do not anticipate that alternative encoder or decoder architectures would yield substantially better performance under the same conditions. We included these experiments in the Supplementary material in the Additional Model Results.

1.

Gut, D. et al. Benchmarking of Deep Architectures for Segmentation of Medical Images. IEEE Trans Med Imaging 41, 3231–3241 (2022).

2.

Isensee, F., Jaeger, P. F., Kohl, S. A. A., Petersen, J. & Maier-Hein, K. H. nnU-Net: a self-configuring method for deep learning-based biomedical image segmentation. Nat Methods 18, 203–211 (2021).

3.

Huang, G., Liu, Z., van der Maaten, L. & Weinberger, K. Q. Densely connected convolutional networks. arXiv [cs.CV] (2016) doi:10.48550/ARXIV.1608.06993.

4.

Zhou, Z., Siddiquee, M. M. R., Tajbakhsh, N. & Liang, J. UNet++: A Nested U-Net Architecture for Medical Image Segmentation. arXiv [cs.CV] (2018) doi:10.48550/ARXIV.1807.10165.

5.

Chen, L.-C., Zhu, Y., Papandreou, G., Schroff, F. & Adam, H. Encoder-decoder with atrous separable convolution for semantic image segmentation. *arXiv [cs.CV]* (2018) doi:10.48550/ARXIV.1802.02611.

6.

Long, J., Shelhamer, E. & Darrell, T. Fully convolutional networks for semantic segmentation. *arXiv [cs.CV]* (2014) doi:10.48550/ARXIV.1411.4038.

Comment R1.2:

A large number of variables and annotations were included in the study in an attempt to improve model performance and explainability (including the profile of three participants selected for each core, demographic profile of the cases included, 32 ontology terminologies, three Gleason patterns, and the presence of technical replicates for each core). While the authors emphasized the accuracy of the annotations by constraining the diversity and complexity of the dataset to just TMA cores and a sample size of 1,015, a formal sample size calculation should be considered to determine the minimum number of cases required to ensure accuracy and reliability. By virtue of the study design, generalizability to clinical practice remains a concern unless a broader cohort encompassing multi-site biopsies or whole slide images is included. At least for validation purposes, an additional cohort comprising these entities could be used to assess performance in predicting only the Gleason score (against the clinical Gleason score available from sign-outs), considering that annotating large slides would be challenging.

REPLY:

We split the response to the comment into two parts. The first (R1.2.1) pertains to the request for a sample size calculation. In the second part of the response (R1.2.2), we go into depth regarding external validation.

R1.2.1 Sample size calculation

We appreciate the reviewer's comment and concerns regarding the sample size. However, we previously decided against a sample size calculation, as our machine learning approaches improve in predictive quality with increasing dataset size, which calls for the use of all available data in order to maximize information extraction. Furthermore, our work is based on retrospective data, therefore, increasing the amount of used data does not burden patients or clinics. We analysed our exclusion criteria and found that they do not introduce any bias (see addition to Supplementary Materials: Data Selection). The retrospective nature of the data and our aim to maximize the amount of accessible data for our machine learning algorithm renders the sample size calculation non-applicable. We clarify the retrospective nature of the data collection in our results (page 5, line 16), but decided against including such a calculation in the paper.

For the reviewer's consideration, we have included an exemplary calculation in the point-by-point, based on references [1] and [2]. However, we would like to reiterate that this calculation should not be performed on retrospectively collected data, contains several limitations and therefore should not be published.

Given the limited literature on each of the explanations and their discriminability, we employed a pixel wise area under receiver operating characteristic curve (AUROC) score in a one-vs-rest scheme, transformed to Cohen's d [1] as the effect size for the sample size estimation [1]:

$$d = \sqrt{2} \cdot \text{erf}^{-1}(2 \cdot \text{AUROC} - 1)$$

, which can then be used to get the estimated sample size with [2]:

$$n = \left(\frac{Z_{1-\alpha/2} + Z_{1-\beta}}{d} \right)^2 \cdot 2$$

Assuming a minimal expected AUROC of 0.65 to be sufficient to ensure reliability, this results in a requirement of 53 samples per explanation class (see Figure 1). Our dataset, consisting of 57 to 729 TMAs per class, exceeds this requirement for all classes, assuring reliability in the classification of these histological patterns.

Figure 1.: Sample Size based on binary AUROC. Number of samples needed depending on the binary AUROC achieved in the class. The red lines indicate the threshold AUROC of 0.65 and the corresponding sample size estimate of $n = 53$.

For future data collection, we also calculated sample size requirements based on the observed pixel wise AUROC achieved by the GleasonXAI on the hard labels of the test set. These calculations yield estimated sample sizes ranging from 4 to 91 samples depending on the class (see Table 1).

All classes except for *Glomeruloid glands* exceed their sample size derived on this analysis. Thus, the AI system can be assumed to yield robust results for those explanations. For *Glomeruloid glands*, for which the sample size remains insufficient, we recommend adding more samples

containing this class before any clinical deployment. For added robustness, we also suggest increasing the number of samples for *Comedonecrosis*, which is close to the minimum threshold. The need for additional targeted sampling of underrepresented classes was also emphasized in the Discussion section of our paper.

Table 1: Sample size per Explanation. AUROC score with 95% confidence intervals (CI) (bootstrap n=1000), the resulting necessary sample size, and the number of TMAs in our dataset in which the explanations occurred. Sample size that was not reached in the dataset in bold.

Explanation	AUROC	95% CI	Calculated Sample Size	Actual Sample Size
Individual glands	0.876	[0.876, 0.876]	6	563
Compressed glands	0.827	[0.827, 0.828]	9	299
Poorly formed glands	0.803	[0.803, 0.803]	11	729
Cribriform glands	0.919	[0.919, 0.920]	4	395
Glomeruloid glands	0.616	[0.614, 0.617]	91	57
Solid group of tumor cells	0.917	[0.917, 0.917]	5	273
Single cells	0.746	[0.744, 0.748]	18	111
Cords	0.940	[0.940, 0.940]	4	222
Comedonecrosis	0.627	[0.626, 0.628]	75	82

1.

Salgado, J. F. Transforming the Area under the Normal Curve (AUC) into Cohen's d, Pearson's r pb , Odds-Ratio, and Natural Log Odds-Ratio: Two Conversion Tables. Eur. J. Psychol. Appl. Leg. Context 10, 35–47 (2018).

2.

Chow, S.-C., Shao, J., Wang, H. & Lokhnygina, Y. Sample Size Calculations in Clinical Research: Third Edition. (Chapman and Hall/CRC, Third edition. | Boca Raton : Taylor & Francis, 2017. | Series: Chapman & Hall/CRC biostatistics series | 'A CRC title, part of the

Taylor & Francis imprint, a member of the Taylor & Francis Group, the academic division of T&F Informa plc.', 2017).

R1.2.2 External Validation:

For the external validation, we created a pipeline applying GleasonXAI to whole slide images using a sliding window approach. To derive a Gleason score from the predicted explanations of the GleasonXAI would introduce a new processing step that would need further optimization. We decided to instead use external datasets with pixel-wise annotation of Gleason patterns (GP3, GP4, GP5). We test our model against these by predicting explanations and remapping them to their respective Gleason pattern.

First, we tested GleasonXAI on the AGGC challenge [1]. The dataset consists of three subsets, with 45, 16, and 67 test images, and 105, 37, and 144 training images, respectively. The first and third subsets contain whole mount images, whereas the second consists of biopsy images. All subsets are scanned with an Akoya scanner. The whole mounts of the third subset are also scanned by up to five additional, different scanners [1]. The challenge contains annotations for GP3, GP4, GP5, normal and stroma tissue. The annotations are not exhaustive, so in the assessment for the challenge, the weighted-average F1-score is calculated on the annotated area only. The F1 scores of the subsets are afterwards combined into a final score in a weighted manner. Since our model doesn't predict normal and stroma tissue, we won't report a final score, but calculate the F1 scores in the whole annotated area for GP3, GP4, and GP5, and for a class representing the union of the remaining classes. Additionally, we applied the GleasonXAI to the whole images to demonstrate the general performance.

For a second dataset, we evaluated GleasonXAI on DiagSet Part A [2], which comprises 425 biopsy WSIs annotated by three pathologists for scan background, tissue background, healthy tissue, artifacts, and Gleason patterns 1–5 within outlined regions. To match our model's outputs, we collapsed these labels into GP3, GP4, GP5, and an "Other" category. Because the original DiagSet evaluation focused on binary cancer-versus-noncancer decisions and patch-level classification, a direct comparison was not possible. Instead, we computed per-class F1 scores over all annotated pixels and applied GleasonXAI to the whole image for qualitative visualization.

To use the images for GleasonXAI, they were reformatted to a common physical pixel side length of 1.392 $\mu\text{m}/\text{px}$. A sliding window approach was applied using a 50% overlap and patch sizes of 512x512px². The patches were recombined using an average merger. Patches without tissue were skipped during inference. For the identification of the region of interest, we tested the inclusion of a patch-wise tissue subtyper trained on colon tissue for the detection of tumorous tissue [3], since GleasonXAI was trained mainly on such tumor tissue, but no benefit was found. The predictions were mapped to Gleason grades, and the final prediction generated by taking the class with the highest probability per pixel. The predictions were evaluated on the tissue area determined through Otsu thresholding as described in our paper. The metrics were determined on the annotated area only.

To test transferability to other domains, and due to the difference in labels, we chose to apply GleasonXAI without further transfer training.

As shown in **Table 1**, the GleasonXAI achieved constantly high F1 scores in GP 3, ranging from 0.667 to 0.722, and notable performance for GP 4 with F1 scores up to 0.704 on the AGGC's whole mount images (AGGC part 1 and 3). It further showed very good performance in the differentiation between tumorous and non-tumorous tissue, as demonstrated by the good F1 scores of the other tissues in the evaluation area. This can also be seen in the visualizations of the results in **Figure 1**, where few areas outside the non-exhaustive ground truth labels are classified with any Gleason pattern, even without further adjustments and pre-selection of tumor tissue. These results highlight the model's robustness in identifying intermediate-grade tumors. In addition, the precision values for GP3 and GP4 were remarkably high (up to 0.960 for GP4 in part 3), suggesting that GleasonXAI is particularly conservative in assigning these labels.

A subanalysis of AGGC part 3 revealed a low difference in performance between the scanners, with the standard deviation in the F1 scores ranging from 0.006 to 0.055 between the results on the scanner subsets, depending on the class, which indicates robustness against changes in scanners.

Figure 1: Predictions on AGGC test set. Examples of the segmentation results of the GleasonXAI on the AGGC data, showing the WSI, the background-adjusted prediction created by the GleasonXAI within and outside of the evaluation area, and the ground truth label within the evaluation area. Panels a) to c) show examples of good agreement between the prediction and the ground truth. Overall, as presented in panels d) to f), there is a tendency for over segmentation of Gleason pattern 5.

We observe pronounced oversegmentation of GP 5. Because this pattern is rare in the training set for the GleasonXAI, we used a macro soft-Dice loss, which averages the Dice error over all classes and therefore gives the minority class the same weight as the majority ones. This equal weighting boosts the gradient signal for GP 5 and was necessary for the model to learn the pattern in our dataset, but it also increases the risk of false-positive GP 5 predictions when the decision threshold or calibration are not perfectly tuned. Under the additional domain shift between the training data and the new cohort, this sensitivity manifests as oversegmentation of GP 5, reflected by high recall but low precision, resulting in a low per-class F1 score. A general decrease in performance was expected, as domain-shifts are a well-known phenomenon in machine learning. To increase the transferability, future work could use foundation models pre-trained on a large variety of images from varying sources and collect more diverse data from more scanners and clinics.

Nonetheless, most of the misclassification is into adjacent classes or tissue not annotated with Gleason patterns 3, 4, or 5. This is similar to the results found in our dataset, and can be further inspected in **Figure 2**, where the row-normalized confusion matrices for pixels of all images of the AGGC dataset is presented. As can be seen, there is almost no misdiagnosis between GP 3 and GP 5.

Figure 2: Confusion matrix for AGGC predictions. Confusion matrices for pixels in the evaluation areas of all images in the AGGC dataset.

Overall, GleasonXAI performs better on the whole mount images (AGGC part 1 and 3) than on biopsy images in AGGC part 2 and DiagSet (see **Table 1**), which can likely be attributed to the fragmented nature of biopsies. Biopsies include more border areas, which often induce more artefacts. They contain a higher variety of tissue types, and are less likely to contain larger sheets of similar tissue compared to whole mount images, thus, they are less similar to TMA cores and therefore more likely to contain out-of-distribution patches. For the use of GleasonXAI on biopsies we therefore recommend training the model on a transfer dataset before use.

In conclusion, even in the absence of domain adaptation, GleasonXAI has demonstrated a strong baseline performance in GP3 and GP4, especially on whole mount images. The AI is usable

without previous tumor tissue detection in the preprocessing, but might benefit from it for the Gleason 5 patterns, if a segmentation rather than a patch classification approach is used. Future improvements on the AI should focus on enhancing performance for rare patterns like GP5, and on the extension of the training data with common artefacts, which will further strengthen the model's diagnostic utility in a non-TMA use case.

We include these results as an additional study in our Supplementary Materials in *Additional Results* and adjust the *Discussion* to better reflect these findings.

Table 1: Results AGGC sets and DiagSet. F1-score, Precision, and Recall achieved by the GleasonXAI on the AGGC test set (Subset 1), AGGC training set (Subset 2) and on the DiagSet for each of the Gleason patterns and the combined class of other annotated tissue types on the tissue pixels in the annotated areas of the images.

	AGGC part 1				AGGC part 2				AGGC part 3				DiagSet			
Metric	Other	GP3	GP4	GP5	Other	GP3	GP4	GP5	Other	GP3	GP4	GP5	Other	GP3	GP4	GP5
Subset 1													Full Set			
F1 score	0.791	0.722	0.622	0.354	0.878	0.667	0.191	0.057	0.708	0.720	0.704	0.025	0.911	0.379	0.446	0.310
Precision	0.698	0.679	0.851	0.284	0.814	0.847	0.295	0.030	0.593	0.756	0.960	0.013	0.916	0.347	0.600	0.211
Recall	0.913	0.771	0.490	0.470	0.953	0.552	0.141	0.614	0.876	0.687	0.556	0.391	0.907	0.417	0.256	0.575
Subset 2																
F1 score	0.705	0.673	0.580	0.304	0.899	0.706	0.441	0.079	0.677	0.779	0.711	0.058				
Precision	0.579	0.690	0.850	0.250	0.857	0.795	0.570	0.042	0.582	0.803	0.910	0.030				
Recall	0.896	0.660	0.440	0.389	0.945	0.635	0.359	0.636	0.807	0.756	0.585	0.701				

1.

Huo, X. et al. A comprehensive AI model development framework for consistent Gleason grading. *Commun Med (Lond)* 4, 84 (2024).

2.

Koziarski, M. et al. DiagSet: a dataset for prostate cancer histopathological image classification. *Sci Rep* 14, 6780 (2024).

3.

Höhn, J. et al. Colorectal cancer risk stratification on histological slides based on survival curves predicted by deep learning. *NPJ Precis Oncol* 7, 98 (2023).

Comment R1.3:

There are a total of seven ontologies in Fig. 4a that were identified by only one pathologist. This raises questions about how the three pathologists for each core were selected, and whether there were significant differences in their professional expertise and profiles. Such variation could introduce potential bias in the annotations.

REPLY:

Thank you for this valuable comment. We have addressed this point by including additional details in the *Methods* section under *Annotation Procedure*:

To minimize potential bias related to individual pathologists' experience, we ensured that the average experience within each group of three pathologists annotating the same dataset was at least 10 years. Pathologists with less than or equal to 5 years of experience were systematically grouped with two highly experienced colleagues (≥ 15 years), ensuring a balanced and robust annotation process. Only in one group did the average annotation experience fall slightly below 10 years, due to the unforeseen dropout of a highly experienced pathologist for whom no suitable replacement could be found.

Comment R1.4:

It is uncertain whether similar results would be expected from large external datasets like TCGA and CPTAC without performing the analysis. While tailored annotations are not available, updated annotation of the existing whole slide images is feasible—at least in terms of Gleason score and pattern.

REPLY:

We understand the need for a further external dataset and an exploration of the transferability of our approach. As our model doesn't contain a prediction of the final Gleason score, the proposed datasets TCGA and CPTAC did not match our requirements for a possible use for evaluation. We therefore applied the GleasonXAI to two other external datasets consisting of whole slide images of different origins.

As demonstrated in our answer to the previous comment Reviewer #1 - Comment 2, a straightforward application of our model to datasets with higher variety, such as whole mounts and biopsies, is possible, but comes with limitations. While we found that the detection of Gleason pattern 3 and 4 was transferable without transfer learning or larger adjustments to the preprocessing, this did not hold for Gleason pattern 5. We therefore would strongly recommend further extension of the training set to the new domain and, as mentioned in the paper, collection of further samples of Gleason pattern 5.

We have added the new analysis in the Supplementary Materials.

New Reviewer #2 who acted as replacement for original reviewer 2
(Remarks to the Author):

Comment R2.1:

In response to the critique regarding software quality, the authors note improvements in documentation and usability, but they do not explicitly address the absence of unit tests or provide assurance of robust versioning practices—both critical for reproducibility and long-term maintainability.

REPLY:

We thank the reviewer for this comment. As our repository is purely meant to enable reproduction of our results, and therefore no future code change is expected, we originally did not include unit tests or versioning.

We again improved our code structure, now providing lock files to guarantee the same python environment, an installable package, passing unit and integration tests for the most important components and improved documentation, as well as versioning of the project, with via tags and a changelog.

Comment R2.2:

Regarding novelty, the response reiterates the work's intended contributions but does not convincingly refute the reviewer's concern that similar ideas around hierarchical modeling and interpretability have been explored previously, leaving the originality of the work ambiguous.

REPLY:

We appreciate the reviewer's comments and the opportunity to further clarify the novelty of our work. In response, we have revised the *Discussion* section to better articulate our contributions.

We fully acknowledge that hierarchical modeling and interpretability are well-established research areas in machine learning. However, the innovation of our study lies in the domain-specific adaptation and the clinical granularity with which we apply these concepts to the task of Gleason grading in histopathology.

Specifically, our work introduces a comprehensive hierarchical framework that captures all 9 ISUP-conformant Gleason patterns, thereby providing detailed and clinically interpretable predictions directly aligned with diagnostic practice. While prior approaches, such as Silva-Rodriguez et al., have incorporated hierarchical detection of a single subpattern [1], our method significantly expands on this by modeling the full spectrum of Gleason morphologies. Moreover, while Explainoma and related methods have explored interpretability in other domains like dermatology [2], these have not been translated or validated in the context of prostate cancer histopathology, which presents distinct morphological and clinical challenges.

Thus, although the underlying concepts of hierarchical learning are not entirely new, their tailored implementation, extension to a full pattern set, and integration with clinical pathology criteria represent a novel contribution to the field. We believe this domain-specific innovation with practical clinical relevance is an important step forward and adds original value to the literature. Additionally, we are the first to collect a dataset of this kind for Gleason scoring, which holds value for future studies in machine learning explainability, but also in assessing the perception of tissue types by pathologists and their discordance.

1.

Silva-Rodríguez, J., Colomer, A., Sales, M. A., Molina, R. & Naranjo, V. Going deeper through the Gleason scoring scale: An automatic end-to-end system for histology prostate grading and cribriform pattern detection. *Comput Methods Programs Biomed* 195, 105637 (2020).

2.

Chanda, T. et al. Dermatologist-like explainable AI enhances trust and confidence in diagnosing melanoma. *Nat Commun* 15, 524 (2024).

Comment R2.3:

Furthermore, in addressing data exclusion and pixel-level disagreements, while the Authors provide an exhaustive breakdown, they offer **limited analytical insight into how these exclusions impact the generalizability or clinical utility of the model.**

REPLY:

We thank the reviewer for noting our efforts in clarifying our exclusions.

A general analysis of the excluded pixels - those for which no consensus was reached - was conducted for the previous review and is documented within the manuscript. This analysis revealed that the majority of the excluded pixels were associated with benign tissue in at least one annotation. Therefore, the primary contribution of these pixels, had they been included, would likely pertain to improving the precision of tissue delineation rather than influencing the final class assignment of the larger tissue structures, which wouldn't hamper clinical utility. Moreover, the overall performance of the hard-label models is comparable to that of the soft-label models. Therefore, it can be concluded that the exclusion of these pixels only has a negligible effect on model performance.

Importantly, the exclusion of these pixels is limited to models trained using hard labels and impacts neither the generalizability nor the clinical utility of GleasonXAI, which is based on soft-label models. As such, the integrity and applicability of the soft-label model remain unaffected by this exclusion.

We experimentally found that the inclusion of images with less than three annotators, due to missing or dropped-out annotators, degraded model performance, as the ground-truth was based on fewer estimates and was therefore overconfident in its representation of pathologists' consensus compared to three-rater-based labels. The resulting decrease in clinical utility validated their removal.

The unwarranted exclusion of images can negatively impact the generalization performance of machine learning models, particularly if the missing data includes important variations or underrepresented classes. To assure that the impact on model generalization is limited and to be able to rule out a systematic bias in the dropped images, we carried out further analyses regarding the grades and explanations on the TMAs without three annotators, which can be found in the following and is also now included in the Supplementary material Data Selection:

Provided Grades:

a)

Combination of Assigned Grades

b)

Combination of Assigned Grades of Dropped Images

c)

Fig.2: Assigned Grade Distribution in dropped images. Distribution of assigned grades a) in the images before dropping images with more than three annotators, b) of the images dropped due to the number of raters and c) the percentage of dropped images per grade combination.

To understand the potential biases introduced by the removal of images that were not fully annotated, we compared the distributions of the Gleason grade combinations provided in the image metadata (e.g. Grade 3+4) between the excluded images and the explanation-annotated dataset before removal. As shown in **Fig. 2**, the majority of dropped images were those assigned a single Gleason grade (e.g. Gleason 3+3, 4+4, or 5+5), which aligns with the most frequently occurring grade combination in the annotated dataset. Overall, the distribution of removed images closely reflects that of the full dataset, as illustrated in **Fig. 2c**, where similar proportions of images were excluded across the various grade combinations. A notable exception is the Gleason Grade 3+5 combination, from which no images were removed. However, due to its infrequent occurrence, this deviation is likely attributable to statistical variability rather than systematic bias.

To evaluate whether the distribution of images across the Gleason Grade combinations changed significantly following the removal of the images, a χ^2 test [1] was conducted at a significance level of $\alpha = 0.05$. The test yielded a p-value of $p = 0.377$, indicating no statistically significant difference between the distributions. Consequently, the null hypothesis was retained, suggesting that the image removal process did not introduce a bias in the distribution of Gleason grades.

Explanation Annotations:

Since images were excluded due to missing annotations, we investigated the explanations provided by raters who did not skip these images. Specifically, we examined whether particular labels were disproportionately represented among the removed images. Our analysis revealed that no explanation was markedly overrepresented in the removals, and the overall distribution of explanation categories in the dropped subset closely mirrored that of the full dataset (see Fig. 3 and Tab. 2). An exception was observed for *Glomeruloid glands*, which compared to their overall occurrence appeared at a slightly higher relative frequency in the removed image. Nonetheless, this minimal increase does not represent a systematic problem.

Fig. 3: Gleason pattern and Explanation occurrences. Number of TMAs with occurrences of a) Gleason patterns and b) explanations in the explanation annotated data and the images dropped due to rater issues.

Tab. 2: Number of TMAs with the given explanation in the dropped and in the overall annotated data, and the ratio of dropped to overall annotated TMAs.

Explanation	TMAs with explanation in dropped data	TMAs with explanation in explanation annotated dataset	Ratio (in %)
Individual glands	83	646	12.85
Compressed glands	42	311	13.50
Poorly formed glands	86	815	10.55
Cribriform glands	46	441	10.43
Glomeruloid glands	12	69	17.39
Solid group of tumor cells	40	313	12.77
Single cells	7	118	5.93
corde	23	245	9.39
comedonecrosis	12	94	12.77

Tab. 3: Number of TMAs with the given Gleason pattern in the dropped and in the overall annotated data, and the ratio of dropped to overall annotated TMAs.

Gleason Pattern	TMAs with Gleason pattern in dropped data	TMAs with Gleason pattern in explanation annotated dataset	Ratio (in %)
3	83	649	12.79
4	98	854	11.48
5	48	376	12.77

1.

Pearson, K. X. On the criterion that a given system of deviations from the probable in the

case of a correlated system of variables is such that it can be reasonably supposed to have arisen from random sampling. Lond. Edinb. Dublin Philos. Mag. J. Sci. 50, 157–175 (1900).

Comment R2.4:

Finally, the justification for introducing the SoftDiceLoss remains largely empirical and lacks rigorous benchmarking or theoretical grounding, leaving open questions about potential overfitting.

REPLY:

The need for the SoftDiceLoss, which is a straightforward extension of the DICE loss, arose through the usage of soft-labels, which were needed to better model the disagreement of the annotators, as hard-labels produced an overconfident estimate of the voting behaviour of the pathologists. Soft-labels have been used extensively before in the deep learning literature for classification tasks in the form of label-smoothing, a regularization technique used in multiple state-of-the-art training procedures [1]. Also in the domain of medical image segmentation, multiple prior works have made efforts to integrate the annotation uncertainty, expressed in the disagreement of annotations, into the training process to better account for this effect, e.g. [2,3,4,5].

In our paper we compare the performance of the hard-label-based DICE loss function with the SoftDiceLoss and observe a slight improvement with using the SoftDiceLoss, that we explain firstly with the accurate depiction of the reviewers uncertainty and secondly with the ability to use pixels where no majority vote exists, and as such under hard-label assumptions cannot be assigned with a label.

We also observe good evaluation results in non-DICE-based metrics and neither signs of overfitting between the training and test results, nor strong differences when trained with or without soft-labels (and therefore with the SoftDiceLoss or just the hard-label based DICE loss).

With the aforementioned results and our knowledge that there has not been an extensive trial-and-error approach to optimize the loss function, but rather a warranted design decision, based on the discordance of annotators, combined with literature of using smoothed/soft labels, we can reassure the reviewer that the loss function is not overfitted to the data.

1.

Müller, R., Kornblith, S. & Hinton, G. When Does Label Smoothing Help? Conference on Neural Information Processing Systems (NeurIPS), 2019

2.

Gros, C., Lemay, A. & Cohen-Adad, J. SoftSeg: Advantages of soft versus binary training for image segmentation. *Medical Image Analysis* **71**, 102038 (2021).

3.

Zhang, J., Zheng, Y. & Shi, Y. A Soft Label Method for Medical Image Segmentation with Multirater Annotations. *Comput Intell Neurosci* **2023**, 1883597 (2023).

4.

Felfeliyan, B. *et al.* Weakly Supervised Medical Image Segmentation With Soft Labels and Noise Robust Loss. in vol. 13644 603–617 (2023).

5.

Silva, J. L. & Oliveira, A. L. Using Soft Labels to Model Uncertainty in Medical Image Segmentation. Preprint at <https://doi.org/10.48550/arXiv.2109.12622> (2021).

Original Reviewer #3 (Remarks to the Author):

We thank the reviewer for reviewing our paper and their continued appreciation of our work.